# Scaling Continual Learning to 300+ Tasks with Bi-Level Routing Mixture-of-Experts

Meng Lou [1]   Yunxiang Fu [1 2]   Yizhou Yu [1 2]

## Abstract

Continual learning, especially class-incremental learning (CIL), on the basis of a pre-trained model (PTM) has garnered substantial research interest in recent years. However, how to effectively learn both discriminative and comprehensive feature representations while maintaining stability and plasticity over very long task sequences remains an open problem. We propose **CaRE**, a scalable **C**ontinual Le**a**rner with efficient Bi-Level **R**outing Mixture-of-**E**xperts (BR-MoE). The core idea of BR-MoE is a bi-level routing mechanism: a router selection stage that dynamically activates relevant task-specific routers, followed by an expert routing phase that dynamically activates and aggregates experts, aiming to inject discriminative and comprehensive representations into every intermediate network layer. On the other hand, we introduce a challenging dataset, OmniBenchmark-1K, for CIL performance evaluation on very long task sequences with hundreds of tasks. Extensive experiments show that CaRE demonstrates leading performance across a variety of datasets and task settings, including commonly used CIL datasets with classical CIL settings (e.g., 5-20 tasks). To the best of our knowledge, CaRE is the first continual learner that scales to very long task sequences (ranging from 100 to over 300 non-overlapping tasks), while outperforming all baselines by a large margin on such task sequences. We hope that this work will inspire further research into continual learning over extremely long task sequences. Code and dataset are publicly released at https://github.com/LMMMEng/CaRE.

[1]School of Computing and Data Science, The University of Hong Kong [2]Center for Embodied Artificial Intelligence and Computer Vision, Shenzhen Loop Area Institute. Correspondence to: Yizhou Yu <yizhouy@acm.org>.

*Proceedings of the $43^{rd}$ International Conference on Machine Learning*, Seoul, South Korea. PMLR 306, 2026. Copyright 2026 by the author(s).

## 1. Introduction

Real-world scenarios often involve streaming data in continually evolving environments (Gomes et al., 2017). Under such circumstances, conventional learning systems generally suffer from catastrophic forgetting, as newly acquired information tends to overwrite historical knowledge (De Lange et al., 2021). To this end, continual learning (CL) (Wang et al., 2024; Yang et al., 2025) has emerged as a promising solution for handling non-stationary data streams while mitigating catastrophic forgetting.

As one of the most challenging settings in CL, class-incremental learning (CIL) (Zhou et al., 2024c) requires a model to continuously learn newly arriving tasks with previously unseen object classes while maintaining its knowledge learned from previously seen ones. Instead of training models from scratch (Li & Hoiem, 2017; Aljundi et al., 2017; Rebuffi et al., 2017; Wu et al., 2019; Hou et al., 2019; Douillard et al., 2020; Yan et al., 2021), recent efforts have leveraged pre-trained models (PTMs) (Zhou et al., 2024a) to exploit their extensive knowledge learned from large-scale datasets such as ImageNet-21K (Deng et al., 2009). PTM-based CIL methods typically adopt parameter-efficient fine-tuning (PEFT) techniques (Hu et al., 2022; Jia et al., 2022; Chen et al., 2022), and can be roughly divided into two categories: prompt-based CIL (Wang et al., 2022b;a; Smith et al., 2023; Jung et al., 2023) and adapter-based CIL (McDonnell et al., 2023; Zhou et al., 2024b; 2025; Gao et al., 2025). In particular, recent works on adapter-based CIL (Sun et al., 2025b; Wu et al., 2025; He et al., 2025; Wang et al., 2025a;b) construct a set of task-specific adapters during continual training and activate appropriate adapters at inference time, achieving promising performance. In this paper, we investigate the following problem with respect to this recent approach: *what properties should the continual learner possess to realize its full potential*?

**Discriminative and Comprehensive Representation Learning.** As there exists a pool of task-specific adapters, it is important to activate the adapter that produces the most discriminative representation for each input sample. This often means identifying the task that most likely includes the class of the input sample, since a task-specific adapter generates feature representations highly discriminative among

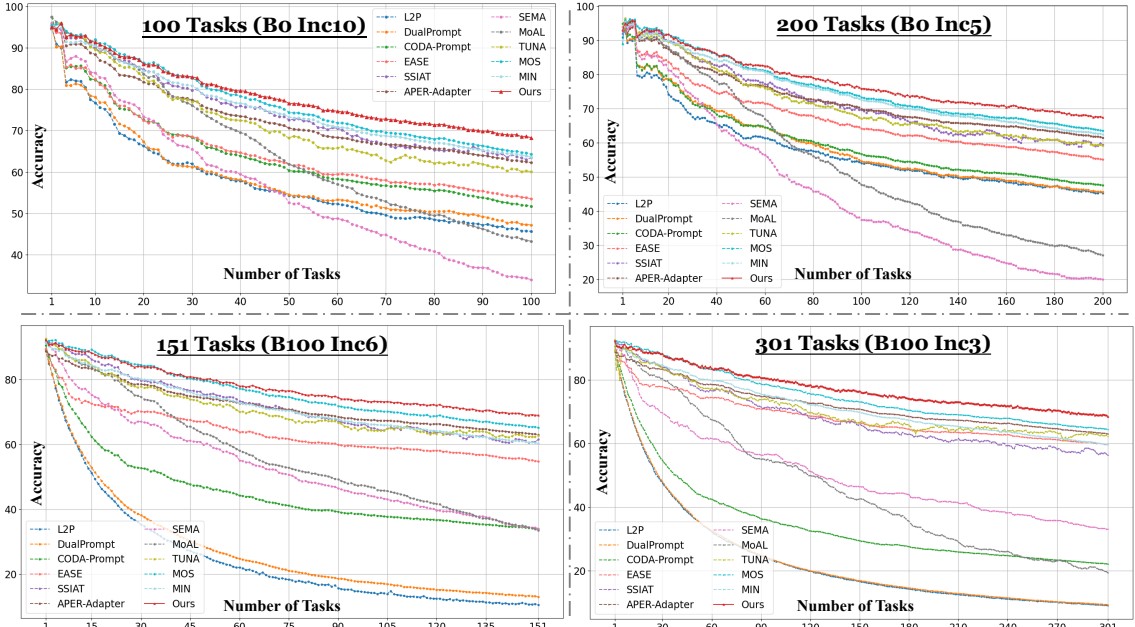

*Figure 1.* Incremental performance comparisons between our CaRE and other representative PTM-based CIL methods on the long-sequence evaluation protocol using the OmniBenchmark-1K dataset. Our method outperforms other baselines by a large margin across a variety of settings. "B-$\mathcal{M}$ Inc-$\mathcal{N}$" denotes the number of base classes ($\mathcal{M}$) and the number of incremental classes ($\mathcal{N}$) per task.

the classes included in its corresponding task. However, a single task only includes a limited number of classes, and being discriminative among them does not necessarily imply a strong discriminative power among other related classes. As the task sequence grows, different tasks may include wider collections of distinct but semantically related classes (e.g., various animal species). How can we make the representation discriminative among them? Existing work along this line typically employs global prompts or adapters derived from all previous tasks (Wang et al., 2022a; Sun et al., 2025b; Wang et al., 2025b). Such coarse-grained strategies are incapable of effectively exploiting fine-grained complementary knowledge. For example, while distinguishing cats from dogs, complementary cues should be primarily drawn from animal-related tasks rather than from unrelated domains such as buildings. Therefore, it is crucial to retrieve and integrate complementary knowledge from relevant historical tasks when learning new tasks. This aligns with human cognition, where the recall of relevant prior knowledge facilitates the acquisition of new information (Tse et al., 2007; Karpicke & Blunt, 2011; van Kesteren et al., 2018).

**Multi-level Local Decisions.** In vision models, as feature representations at different depths have different levels of abstraction (Yan et al., 2015; Lou et al., 2025a), a continual learner should possess the ability to make local decisions at each intermediate network layer to selectively incorporate both discriminative and complementary historical knowledge. Such a local decision strategy injects customized knowledge retrieval capabilities into every network layer.

**Performance Evaluation on Long Task Sequences.** In real-world applications, a continual learner should be able to adapt to scenarios where the number of tasks continually increases and reaches a large number. However, previous studies have primarily been validated on a limited number of tasks (e.g., 20 tasks), leaving it unclear how these approaches would perform on longer task sequences. This is largely because common CIL datasets suffer from a limited number of classes. For instance, CIFAR-100 (Krizhevsky et al., 2009), a widely used benchmark, contains 100 classes only, making it unsuitable for long-sequence evaluations, since partitioning it into 100 tasks reduces each to a trivial single-class learning problem. Although the ImageNet dataset appears to be an option, it is not ideal for evaluating PTM-based CIL methods, which typically utilize weights pre-trained on the ImageNet dataset, leading to biased results. Hence, there is a clear need for a more challenging dataset that enables scalable CIL assessments under long task sequences.

Given the preceding considerations, we propose **CaRE**, a scalable **Ca**ntinual Lea**R**ner featuring a novel Bi-Level **R**outing Mixture-of-**E**xperts (BR-MoE) mechanism. As the core of CaRE, BR-MoE learns a triplet of parameter-efficient, task-specific components at each incremental step: a class perceptron, a router network, and an adapter. As shown in Figure 2, BR-MoE adopts a bi-level routing mechanism comprising a dynamic router selection stage and a subsequent dynamic expert routing stage. In the first stage, an input feature is fed into every task-specific class percep-

tron to produce semantic guidance, which is then used to select Top-**M** task-specific router networks. In the second stage, each selected router network generates dynamic gating coefficients, the Top-**K** of which activate and aggregate the corresponding task-specific adapter experts, yielding a refined output feature. This design encourages the model to not only maintain task-specific knowledge, but also dynamically retrieve and reuse relevant knowledge from all learned tasks, thereby producing both discriminative and comprehensive feature representations. By equipping each intermediate layer with BR-MoE, the continual learner can dynamically make local routing decisions that improve the overall performance during incremental adaptation.

To address the absence of a suitable benchmark for evaluating CIL methods on long task sequences, we introduce a challenging dataset named OmniBenchmark-1K, curated from the OmniBenchmark-V2 dataset (Zhang et al., 2022). OmniBenchmark-1K contains 1,000 classes with around 190,000 images spanning 21 visual realms, facilitating comprehensive long-sequence evaluations.

We evaluate CaRE through extensive experiments on a variety of datasets. As shown in Figure 1, CaRE delivers impressive performance improvements over other strong PTM-based CIL methods in long-sequence evaluations using OmniBenchmark-1K (from 100 to 301 tasks). For example, at 100 tasks, CaRE surpasses strong baselines such as TUNA (Wang et al., 2025b) by 8.23% in last accuracy ($\mathcal{A}_\mathcal{B}$). At 151 tasks, our method outperforms MIN (Jiang et al., 2025) by 8.68% in $\mathcal{A}_\mathcal{B}$. At 200 tasks, CaRE exceeds APER-Adapter (Zhou et al., 2025) by 5.93% in $\mathcal{A}_\mathcal{B}$. Even when given a very long sequence of 301 tasks, CaRE still yields significant gains over all considered baselines. Meanwhile, as shown in Table 3, CaRE also retains a clear advantage on several classical datasets such as ImageNet-R (Hendrycks et al., 2021a) and ImageNet-A (Hendrycks et al., 2021b) in short-sequence settings (e.g., 5-20 tasks). We hope that both the CaRE continual learner and the OmniBenchmark-1K dataset will help advance research in the CL community.

## 2. Related Work

**Class-Incremental Learning (CIL)** has witnessed remarkable progress in recent years (Zhou et al., 2024c). Prevailing methods can be summarized along three main lines: regularization-based (Li & Hoiem, 2017; Aljundi et al., 2017; Hou et al., 2019; Douillard et al., 2020; Ashok et al., 2022; Wen et al., 2024), replay-based (Lopez-Paz & Ranzato, 2017; Riemer et al., 2019; Wu et al., 2019; Chaudhry et al., 2019; Liu et al., 2021; Shin et al., 2017; Van de Ven et al., 2020; Zhu et al., 2021), and optimization-based methods (Farajtabar et al., 2020; Saha et al., 2021; Lu et al., 2024). Recently, CIL with pre-trained models (PTMs) has emerged as a prospective direction, as the powerful prior knowledge embedded in PTMs can effectively mit-

igate catastrophic forgetting and improve overall performance (Zhou et al., 2024a). For instance, L2P (Wang et al., 2022b) introduces a learnable prompt pool and learns to retrieve task-specific prompts. Subsequent works such as DualPrompt (Wang et al., 2022a), DAP (Jung et al., 2023), and CODA-Prompt (Smith et al., 2023) further enhance the effectiveness of prompt tuning in CIL. APER (Zhou et al., 2025) demonstrates that a simple shared adapter with a prototype-based classifier can achieve promising performance. EASE (Zhou et al., 2024b) constructs task-specific subspaces by incrementally tuning adapters. MOS (Sun et al., 2025b) improves retrieval accuracy with adapter merging and a self-refined mechanism. TUNA (Wang et al., 2025b) coordinates generic and task-specific adapters during inference. Recently, MIN (Jiang et al., 2025) learns beneficial noise to counteract parameter drift during the incremental learning stage. This paper's contributions can be summarized in the following aspects. First, our CaRE enhances the dynamic modeling capacity of every network layer, encapsulating powerful feature representations into the continual learner. Second, CaRE is the first piece of work to tackle the challenge of scaling CIL to very long task sequences (e.g., over 300 non-overlapping tasks), whereas previous work has largely been confined to short-sequence evaluations (e.g., from 5 to 20 tasks).

**Mixture-of-Experts (MoE)** has recently emerged as a powerful architecture (Rajbhandari et al., 2022; Dai et al., 2024; Cai et al., 2025). The core idea of combining multiple specialized experts through a dynamic gating mechanism has inspired some CL methods. For instance, MoE-Adapter (Yu et al., 2024) trains a dedicated router along with a set of experts for each task on top of a pre-trained vision-language model (Radford et al., 2021). MoE-Adapter++ (Yu et al., 2025) further enhances this design with an expert-expansion controller and a latent embedding auto-selector. DCE (Li et al., 2025) proposes frequency-aware collaborative experts for domain-incremental learning. SEMA (Wang et al., 2025a) presents a self-expansion CIL approach, which automatically decides whether to reuse existing adapters or add new ones. In contrast, our BR-MoE introduces a bi-level routing mechanism with more comprehensive relevant knowledge retrieval and aggregation at every network layer, demonstrating robust performance.

## 3. Method

### 3.1. Preliminaries

Let $\{\mathcal{D}^t\}_{t=1}^{\mathcal{T}}$ denote the datasets for a set of $\mathcal{T}$ tasks. In the dataset $\mathcal{D}^t = \{(x_i^t, y_i^t)\}_{i=1}^{n^t}$ for task $t$, there are $n^t$ input samples and each sample $x_i^t$ is paired with a corresponding label $y_i^t \in G^t$, where $G^t$ denotes the label set for task $t$. The label sets for any two tasks ($t$ and $t'$) are non-overlapping, i.e., $G^t \cap G^{t'} = \emptyset$. The learning objective is to find an

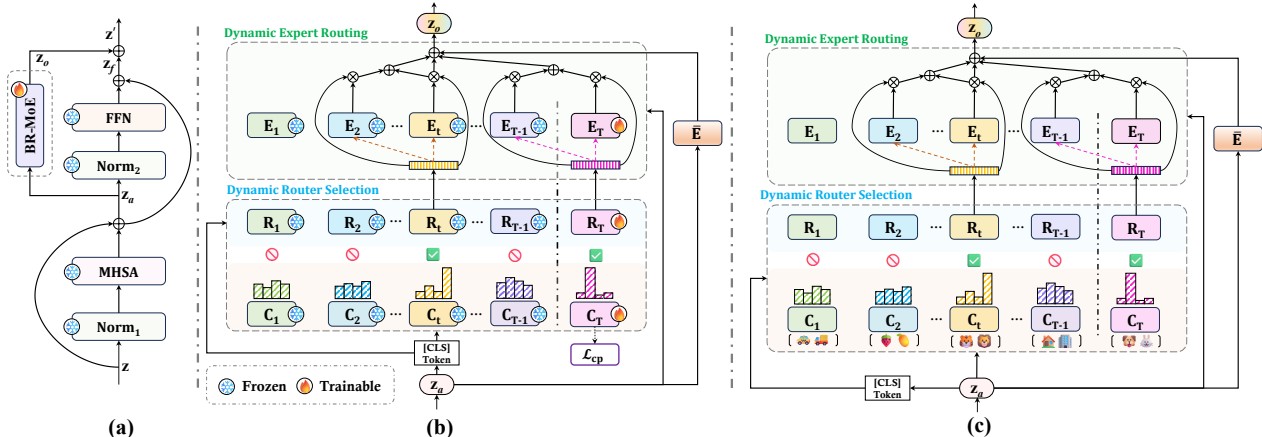

*Figure 2.* The workflow of the proposed BR-MoE. **(a)** The network building block equipped with our BR-MoE. **(b)** Training and **(c)** inference pipelines of BR-MoE.

optimal model at task $t$, denoted as $f^t : \mathcal{X} \to \mathbb{R}^{c^t}$, where $\mathcal{X}$ represents the input space and $c^t = |\cup_{j=1}^t G^j|$ represents the total number of classes learned up to task $t$. In this work, the model $f^t$ is built upon a PTM, and defined as $f^t(x) = \mathbf{W_t}^\top \phi^t(x)$, where $\phi^t : \mathcal{X} \to \mathbb{R}^d$ is a feature encoder consisting of a frozen PTM and parameter-efficient modules learned up to task $t$. The linear classifier $\mathbf{W_t} \in \mathbb{R}^{d \times c^t}$ is a concatenation of $t$ weight matrices, i.e., $\mathbf{W_t} = [\mathbf{w^1}, \mathbf{w^2}, ..., \mathbf{w^t}]$, where $\mathbf{w^t} \in \mathbb{R}^{d \times |G^t|}$ represents the task-specific weight matrix for task $t$. When the model is trained on task $t$, all parameters learned from the previous $t-1$ tasks remain frozen.

### 3.2. Overall Architecture

As illustrated in Figure 2 (a), the proposed CaRE is built upon a pre-trained ViT (Dosovitskiy et al., 2021). The core of our framework is an efficient Bi-Level Routing Mixture-of-Expert (BR-MoE) module, which is seamlessly integrated into every ViT building block. Following Adapt-Former (Chen et al., 2022), the forward process within a building block equipped with BR-MoE is formulated as follows:

$$\begin{aligned} \mathbf{z}_a &= \text{MHSA}(\text{Norm}_1(\mathbf{z})) + \mathbf{z}, \\ \mathbf{z}_f &= \text{FFN}(\text{Norm}_2(\mathbf{z}_a)) + \mathbf{z}_a, \\ \mathbf{z}' &= \text{BR-MoE}(\mathbf{z}_a) + \mathbf{z}_f, \end{aligned} \quad (1)$$

where MHSA and FFN refer to multi-head self-attention and feedforward network, respectively, while $\mathbf{z}$ and $\mathbf{z}'$ refer to the input and output features. During incremental training, only the components and parameters of the BR-MoE modules are updated to learn new tasks. The classification loss follows the angular penalty function (Peng et al., 2022):

$$\mathcal{L}_{\text{cls}} = -\frac{1}{n^t} \sum_{i=1}^{n^t} \log \frac{\exp(\tau \cos(\theta_i^{y_i^t}))}{\sum_{j=1}^{|G^t|} \exp(\tau \cos(\theta_i^j))}, \quad (2)$$

where $\cos(\theta_i^j) = \frac{w_j^t \cdot \phi^t(x_i^t)}{\|w_j^t\| \|\phi^t(x_i^t)\|}$ denotes the cosine distance

between class $j$ in task $t$ and the feature representation of input sample $x_i^t$, $y_i^t$ is the ground-truth class label of $x_i^t$, $w_j^t$ is the weight vector associated with class $j$ in the weight matrix $\mathbf{w^t}$ for task $t$, and $\tau$ is a scaling factor fixed to 20 following (Tan et al., 2024; Wang et al., 2025b).

### 3.3. Bi-Level Routing Mixture-of-Experts

**Overview.** Every BR-MoE module contains a set of triplet components, $\{(\mathbf{C}_t, \mathbf{R}_t, \mathbf{E}_t)\}_{t=1}^T$, where $\mathbf{C}_t$ is a class perceptron, $\mathbf{R}_t$ is a router network, and $\mathbf{E}_t$ is an expert. There is one triplet associated with each of the $T$ tasks. For a given input feature $\mathbf{z}_a \in \mathbb{R}^{d \times l}$ (where $d$ and $l$ denote the channel and spatial dimensions, respectively), an expert is a parameter-efficient module for feature transformation. We employ two types of experts: a task-specific expert $\mathbf{E}_t$, which is tailored for features pertinent to its associated task $t$, and a shared expert $\bar{\mathbf{E}}$, which encodes cross-task knowledge accumulated from all existing tasks. After learning task $t$, a BR-MoE module contains $t$ task-specific experts and one shared expert. Each expert is implemented as an Adapter module (Chen et al., 2022). A router network $\mathbf{R}_t$ associated with task $t$ comprises a linear layer, $\eta^t \in \mathbb{R}^{d \times t}$, followed by a softmax operation. It projects the [CLS] token in $\mathbf{z}_a$ onto the task-specific experts learned up to task $t$, producing a set of scalar gating scores for those experts. These scores enable *dynamic expert routing*: the Top-**K** task-specific experts with the highest gating scores are activated and aggregated to exploit relevant knowledge, while the shared expert is always activated to further enrich the representation with cross-task knowledge. A class perceptron ($\mathbf{C}_t$) associated with task $t$ generates semantic guidance by extracting class-level discriminative information from $\mathbf{z}_a$. Class perceptrons perform *dynamic router selection* by deciding which Top-**M** router networks are most appropriate for the current input. Specifically, $\mathbf{C}_t$ is implemented as a linear layer, $\rho^t \in \mathbb{R}^{d \times |G_t|}$, mapping the [CLS] token to a set of classification logits for the classes in task $t$. Router

networks are ranked according to the entropy of their logits.

A BR-MoE module dynamically aggregates relevant knowledge from learned tasks through a bi-level process, which first selects multiple most related routers, each of which then activates multiple complementary experts, while a shared expert with consolidated knowledge from all tasks further enriches the feature representation.

**Dynamic Router Selection** aims to dynamically identify the most semantically relevant knowledge for the current input. The core mechanism involves dynamically inferring the most probable task identities and their associated routers in every network layer for a given input sample. Suppose T tasks have been learned or the T-th task is being learned. For a given input feature $\mathbf{z}_a$ of a BR-MoE module, the [CLS] token of $\mathbf{z}_a$, $\mathbf{z}_a^{\text{[CLS]}}$, is fed to every class perceptron in $\{\mathbf{C}_t\}_{t=1}^{\text{T}}$, producing a set of classification logits:

$$\mathbf{s}_t = \text{Softmax}(\mathbf{C}_t(\mathbf{z}_a^{\text{[CLS]}})), \quad \forall t \in \{1, 2, \ldots, \text{T}\}, \quad (3)$$

where $\mathbf{s}_t \in \mathbb{R}^{|G^t|}$ denotes the probability distribution for the classes in task $t$. We further calculate the entropy of the logits produced by every class perceptron as follows:

$$\mathcal{H}_t = -\sum_{j=1}^{|G^t|} \mathbf{s}_t^{(j)} \log(\mathbf{s}_t^{(j)}), \quad \forall t \in \{1, 2, \ldots, \text{T}\}, \quad (4)$$

where $\mathbf{s}_t^{(j)}$ denotes the $j$-th element of $\mathbf{s}_t$. A lower entropy indicates a higher confidence that the input is a sample from one of the classes in the corresponding task. Hence, the router networks paired with the Top-**M** class perceptrons with the smallest entropy values are selected. During training, the router network corresponding to the latest task ($\mathbf{R}_\text{T}$) is always activated, while the remaining **M**−1 routers are selected dynamically according to their entropy values (Figure 2 (b)). During inference, all **M** routers are dynamically chosen in an entropy-driven manner (Figure 2 (c)).

**Dynamic Expert Routing** performs fine-grained feature adaptation once the Top-**M** router networks have been selected. Consider a simple example with **M**=2, where two routers $\{\mathbf{R}_t, \mathbf{R}_\text{T}\}$ have been activated. In practice, $\mathbf{z}_a^{\text{[CLS]}}$ is fed into $\mathbf{R}_\text{t}$, generating a gating vector for the first t experts. Likewise, $\mathbf{R}_\text{T}$ generates another gating vector for the first T experts. To focus on the most relevant knowledge, for each selected router, we only activate the Top-**K** experts with the largest gating scores, which are re-normalized through the softmax operator. Take a simple example of **K**=2, and suppose $\mathbf{R}_\text{t}$ produces Top-2 gating scores $\{a_2, a_t\}$, which correspond to adapters $\{\mathbf{E}_2, \mathbf{E}_t\}$. Meanwhile, suppose $\mathbf{R}_\text{T}$ produces Top-2 gating scores $\{b_{\text{T-1}}, b_\text{T}\}$ for $\{\mathbf{E}_{\text{T-1}}, \mathbf{E}_\text{T}\}$. The resulting feature is calculated as follows:

$$\begin{aligned} \mathbf{z_1} &= a_2\mathbf{E}_2(\mathbf{z}_a) + a_\text{t}\mathbf{E}_\text{t}(\mathbf{z}_a) \\ \mathbf{z_2} &= b_{\text{T-1}}\mathbf{E}_{\text{T-1}}(\mathbf{z}_a) + b_\text{T}\mathbf{E}_\text{T}(\mathbf{z}_a) \qquad (5) \\ \mathbf{z}_r &= \mathbf{z_1} + \mathbf{z_2} \end{aligned}$$

Meanwhile, we introduce a shared expert ($\bar{\mathbf{E}}$) inspired by DeepSeekMoE (Dai et al., 2024). $\bar{\mathbf{E}}$ is implemented as a momentum-based adapter, which is fully trained on the initial task and updated via EMA (Polyak & Juditsky, 1992) for all subsequent tasks:

$$\delta_s \leftarrow \mu\delta_s + (1 - \mu)\delta_t, \qquad (6)$$

where $\delta_s$ represents the parameters of the shared expert, $\delta_t$ represents the parameters of an adapter solely trained on a new task $t$, and $\mu$ is the momentum coefficient (e.g., $\mu$=0.999). Note that there is only one shared expert, which is reused across all learned tasks. The final output $\mathbf{z}_o \in \mathbb{R}^{d \times l}$ of BR-MoE is computed as:

$$\mathbf{z}_o = \text{BR-MoE}(\mathbf{z}_a) = \mathbf{z}_r + \bar{\mathbf{E}}(\mathbf{z}_a) \qquad (7)$$

By default, we set **M**=2 and **K**=3, while a regular adapter and the shared adapter are configured with 16 and 64 bottleneck channels, respectively. Additional configurations are discussed in the experimental section.

**Training Objectives**. When a new task $t$ arrives, our framework learns a triplet of new components $(\mathbf{C}_t, \mathbf{R}_t, \mathbf{E}_t)$ within every BR-MoE module while freezing all parameters learned from previous tasks. To ensure that the class perceptron ($\mathbf{C}_t$) produces accurate classification logits, thereby generating reasonable entropy, $\mathbf{s}_t$ is supervised with its own classification loss, similar to Equation 2. However, compared to final-layer representations, features at intermediate or shallow layers are typically less discriminative because high-level semantic abstractions may not have sufficiently developed. To learn more robust semantic guidance, we introduce a KL divergence loss $\mathcal{L}_{\text{KL}}^\ell$ between $\mathbf{s}_t \in \mathbb{R}^{|G^t|}$ and final-layer softmax probabilities $p_t \in \mathbb{R}^{|G^t|}$ for task $t$, aiming to mimic high-level representations directly. The final loss for the class perceptron at the $\ell$-th layer is:

$$\mathcal{L}_{\text{cp}}^\ell = \mathcal{L}_{\text{cls}}^\ell + \mathcal{L}_{\text{KL}}^\ell \qquad (8)$$

For training stability, we average $\mathcal{L}_{\text{cp}}^\ell$ across all $L$ layers and scale it by a factor $\lambda$ (set to 1 by default), which is then combined with the main classification loss in Equation 2 to form the overall training objective of CaRE:

$$\mathcal{L} = \mathcal{L}_{\text{cls}} + \lambda\frac{1}{L}\sum_{\ell=1}^{L}\mathcal{L}_{\text{cp}}^\ell \qquad (9)$$

That is, in addition to the supervision applied to the classifier at the final layer, the class perceptron at each intermediate layer receives direct supervision as well. This endows every BR-MoE module to develop a local decision-making ability based on the semantic abstraction at its own layer, enabling customized knowledge retrieval.

### 3.4. Discussions of BR-MoE

**Why Entropy for Router Selection?** In information theory, entropy is a fundamental measure of uncertainty, quan-

tifying the expected information content of a probability distribution. Therefore, entropy can capture the prediction uncertainty of each task-specific class perceptron by evaluating its output distribution, thereby effectively identifying inputs that may originate from unrelated tasks. By ranking class perceptrons in an ascending order of entropy values, we obtain a prioritized list of router networks, from the router associated with the most likely task to those with higher uncertainty. This strategy provides a robust foundation for the subsequent multi-router selection mechanism. Extensive empirical results confirm its superior robustness over alternatives (Section 5.3).

**Why Activate Multiple Router Networks?** As presented earlier, router networks are selected according to entropy-based ranking. An input is more likely sampled from a class included in tasks associated with lower entropy, and the routers associated with such tasks produce discriminative representations for the input. Nevertheless, every task is associated with a distinct router, and the representation produced by this router is most discriminative among the classes included in the task. To make the representation discriminative among a wider collection of classes, our method activates multiple top-ranked routers, which produce complementary features that make the representation more comprehensive. This design aligns with the discussion in Section 1. Our experiments in Appendix A.2 demonstrate the effectiveness of activating multiple router networks.

**Learning to Utilize Historical Knowledge.** Our design explicitly forces the model to engage prior knowledge at each network layer when new tasks are learned. At the router level, the dynamic selection of historical routers (besides the current one) ensures that the representations learned for the new task are compatible with the learned gating patterns of related past routers. At the adapter (expert) level, the gating mechanism dynamically retrieves and composes features from frozen historical adapters, directly reusing their encoded knowledge to enhance feature representations. Meanwhile, the shared expert covers the knowledge of all learned tasks. As a result, the learned representations are both comprehensive and discriminative, giving rise to robust continual learning performance.

## 4. A Benchmark for Long Task Sequence Class-Incremental Learning

To enable scalable evaluation of diverse CIL algorithms on very long task sequences, we construct a new benchmark dubbed OmniBenchmark-1K, by curating a subset of 1,000 classes from OmniBenchmark-V2 (Zhang et al., 2022). The original OmniBenchmark-V2 organizes categories into multiple thematic realms (e.g., birds, foods, activities) and removes duplicates that overlap with potential pre-training datasets, including ImageNet-21K (Deng et al., 2009). To

ensure diversity, we sample classes in a roughly balanced manner across these realms. Specifically, for the training set, we first collect all candidate classes containing at least 100 images per realm, then randomly select an approximately equal number of classes from each realm (using a fixed random seed of 1993) to form the dataset. For the test set, we directly extract the corresponding samples for the selected classes from the original test portion, as its image distribution is approximately uniform.

The resulting OmniBenchmark-1K training set comprises 1,000 classes covering all realms of the original dataset, with a total of 188,569 images, including 168,718 training images (an average of 169 per class). The largest class contains 403 samples, while the smallest contains 100. The test set contains 19,849 images, averaging 19 per class, with per-class image counts ranging from 17 to 20. For reference, the commonly used ImageNet-R dataset (Hendrycks et al., 2021a) consists of 200 classes, with 24,000 training images (120 per class on average, ranging from 38 to 349) and 6,000 test images (30 per class on average, ranging from 7 to 81). Additionally, although the OmniBenchmark-V1 has been widely adopted in many CIL works, such as (Zhou et al., 2025; Gao et al., 2025; Sun et al., 2025b; Jiang et al., 2025), these works typically utilize a very small subset of only 300 classes, while V1 contains more low-quality images compared with V2.

In summary, OmniBenchmark-1K not only provides a larger number of classes but also ensures that each class has a sufficient number of training samples to mitigate overfitting in CIL methods, while covering a wide range of complex visual realms. We believe this benchmark offers a challenging and scalable testbed for long-sequence CIL evaluations.

## 5. Experiments

### 5.1. Long Task Sequence Evaluations

**Datasets.** To evaluate CIL methods on long task sequences, we conduct extensive experiments using the proposed OmniBenchmark-1K dataset. To comprehensively assess task scalability, we evaluate performance under multiple configurations denoted as "B-$\mathcal{M}$ Inc-$\mathcal{N}$" (where $\mathcal{M}$ is the number of classes in the first task and $\mathcal{N}$ in each subsequent one): 100 tasks (B0 Inc10), 200 tasks (B0 Inc5), 151 tasks (B100 Inc6), and 301 tasks (B100 Inc3). We also conduct experiments on four standard CIL benchmarks: OmniBenchmark-V1 (Zhang et al., 2022), ObjectNet (Barbu et al., 2019), ImageNet-R (Hendrycks et al., 2021a), and ImageNet-A (Hendrycks et al., 2021b). Given their limited class count, their CIL task sequences are shorter, ranging from 50 to 60 tasks. Following prior works (Wang et al., 2022b; Zhou et al., 2024b; 2025), class order is obtained using a random seed of 1993, while average accuracy ($\bar{\mathcal{A}}$)

*Table 1.* Comparison of average and last accuracy on very long task sequences using the OmniBenchmark-1K dataset.

| Method | $\bar{\mathcal{A}}$ | $\mathcal{A}_\mathcal{B}$ | $\bar{\mathcal{A}}$ | $\mathcal{A}_\mathcal{B}$ |
| | 100 Tasks (B0 Inc10) | | 200 Tasks (B0 Inc5) | |
|---|---|---|---|---|
| L2P | 60.91 | 48.87 | 57.53 | 45.25 |
| DualPrompt | 62.18 | 49.45 | 59.13 | 45.62 |
| CODA-Prompt | 64.16 | 51.75 | 60.22 | 47.56 |
| EASE | 65.00 | 53.54 | 67.23 | 55.13 |
| SSIAT | 74.81 | 63.45 | 71.97 | 59.43 |
| APER-Adapter | 73.23 | 62.24 | 72.49 | 61.53 |
| SEMA | 56.55 | 33.96 | 45.06 | 19.95 |
| MoAL | 63.29 | 41.55 | 53.72 | 27.03 |
| TUNA | 71.93 | 60.04 | 71.45 | 59.14 |
| MOS | 76.80 | 64.27 | 75.92 | 63.51 |
| MIN | 75.46 | 63.60 | 74.94 | 62.50 |
| **CaRE (Ours)** | **78.54** | **68.27** | **78.00** | **67.46** |

| Method | 151 Tasks (B100 Inc6) | | 301 Tasks (B100 Inc3) | |
|---|---|---|---|---|
| L2P | 24.94 | 10.49 | 23.07 | 9.03 |
| DualPrompt | 27.57 | 12.90 | 23.41 | 9.30 |
| CODA-Prompt | 45.43 | 34.01 | 34.77 | 22.11 |
| EASE | 63.78 | 54.76 | 68.15 | 59.54 |
| SSIAT | 71.99 | 61.55 | 68.15 | 56.26 |
| APER-Adapter | 72.10 | 62.99 | 72.08 | 62.99 |
| SEMA | 53.44 | 34.00 | 50.16 | 33.01 |
| MoAL | 56.74 | 33.47 | 46.05 | 19.33 |
| TUNA | 70.80 | 62.77 | 70.17 | 62.21 |
| MOS | 76.01 | 65.20 | 74.91 | 64.37 |
| MIN | 71.74 | 60.33 | 71.37 | 59.63 |
| **CaRE (Ours)** | **77.65** | **69.01** | **77.18** | **68.51** |

*Table 2.* Comparison of average and last accuracy on long task sequences.

| Method | $\bar{\mathcal{A}}$ | $\mathcal{A}_\mathcal{B}$ | $\bar{\mathcal{A}}$ | $\mathcal{A}_\mathcal{B}$ |
| | OmniBenchmark-V1 60 Tasks (B0 Inc5) | | ObjectNet 50 Tasks (B0 Inc4) | |
|---|---|---|---|---|
| L2P | 66.22 | 56.02 | 57.76 | 45.13 |
| DualPrompt | 66.46 | 55.91 | 54.37 | 41.13 |
| CODA-Prompt | 61.58 | 57.06 | 50.39 | 39.38 |
| EASE | 73.06 | 65.10 | 68.65 | 54.45 |
| SSIAT | 82.10 | 72.36 | 74.07 | 62.19 |
| APER-Adapter | 79.95 | 72.43 | 69.07 | 56.23 |
| SEMA | 69.52 | 53.75 | 61.02 | 46.38 |
| MoAL | 77.66 | 60.40 | 62.42 | 40.27 |
| TUNA | 79.35 | 69.39 | 72.56 | 58.37 |
| MOS | **85.31** | 77.16 | 63.70 | 49.02 |
| MIN | 84.65 | 76.02 | 71.96 | 58.92 |
| **CaRE (Ours)** | 84.74 | **77.69** | **76.88** | **65.13** |

| Method | ImageNet-R 50 Tasks (B0 Inc4) | | ImageNet-A 50 Tasks (B0 Inc4) | |
|---|---|---|---|---|
| L2P | 57.86 | 48.33 | 49.89 | 36.41 |
| DualPrompt | 55.38 | 47.40 | 43.85 | 29.95 |
| CODA-Prompt | 55.82 | 45.43 | 38.24 | 26.60 |
| EASE | 76.51 | 68.67 | 59.86 | 47.53 |
| SSIAT | 79.63 | 74.47 | 61.31 | 49.11 |
| APER-Adapter | 69.29 | 61.12 | 61.29 | 48.58 |
| SEMA | 67.80 | 59.32 | 52.99 | 40.68 |
| MoAL | 68.81 | 52.32 | 54.46 | 33.57 |
| TUNA | 80.93 | 75.48 | 67.82 | 58.72 |
| MOS | 75.22 | 67.18 | 63.72 | 51.22 |
| MIN | 82.26 | 75.72 | 66.15 | 57.08 |
| **CaRE (Ours)** | **82.92** | **76.98** | **69.85** | **59.91** |

and last accuracy ($\mathcal{A}_B$) are chosen as performance metrics. Due to limited space, this section only shows a subset of the results, with more results given in the Appendix.

**Training settings.** For fair comparisons, all experiments are conducted using the LAMDA-PILOT codebase (Sun et al., 2025a) on a single NVIDIA H800 GPU. Following common practice in PTM-based CIL (Wang et al., 2022b; Zhou et al., 2025), we employ ViT-B/16-IN21K (Dosovitskiy et al., 2021) as the PTM, while the results obtained using other pre-trained weights are provided in Appendix A.1. During training, we optimize the model with SGD (momentum=0.9 and weight decay=5e-4), use a batch size of 16, and train for 20 epochs per task. The learning rate is initialized to 0.01 and follows a cosine annealing schedule. We perform multiple runs and report average results. The standard deviation is omitted due to its negligible magnitude.

**Baselines.** Our CaRE is compared with numerous representative CIL baselines: L2P (Wang et al., 2022b), DualPrompt (Wang et al., 2022a), CODA-Prompt (Smith et al., 2023), EASE (Zhou et al., 2024b), SSIAT (Tan et al., 2024), APER-Adapter (Zhou et al., 2025), SEMA (Wang et al., 2025a), MoAL (Gao et al., 2025), TUNA (Wang et al., 2025b), MOS (Sun et al., 2025b), and MIN (Jiang et al., 2025). We use the official implementation of each baseline and adhere to the same experimental settings described above to ensure a fair comparison.

**Results.** As listed in Table 1, our proposed CaRE demonstrates significant performance advantages over all com-

pared baselines under diverse long-sequence evaluation settings. Specifically, in the 100-task setting (B0 Inc10), CaRE surpasses MIN and MOS by 4.67% and 4% in last accuracy ($\mathcal{A}_B$), respectively. When scaling to 200 tasks (B0 Inc5), CaRE outperforms TUNA by a notable margin of 8.32% in $\mathcal{A}_B$. Under large base-class configurations, CaRE maintains strong performance, achieving a 6.02% improvement in $\mathcal{A}_B$ in the 151-task setting (B100 Inc6) compared with APER-Adapter. Most notably, when extended to an exceptionally long sequence of 301 tasks (B100 Inc3), CaRE retains a clear performance advantage, exceeding all baselines by a substantial margin. The complete incremental learning curves in Figure 1 further confirm that CaRE delivers the most stable learning trajectory.

A crucial observation is that some advanced methods (e.g., SEMA and MoAL) exhibit significant performance degradation in long-sequence evaluations. Taking MoAL as an example, Figure 1 reveals that while it maintains competitive performance during the early sessions (about the first 20 tasks), its accuracy declines dramatically as the task sequence lengthens. This suggests that while some existing methods achieve promising performance in short-sequence settings, they struggle to scale to hundreds of tasks.

On the other hand, CaRE consistently outperforms other methods on established benchmarks with moderately long task sequences, as shown in Table 2. On OmniBenchmark-V1 (60 tasks) and ObjectNet (50 tasks), CaRE surpasses EASE by significant margins of 12.59% and 10.68% in

*Table 3.* Comparison of average and last accuracy on short task sequences.

| Method | CIFAR-100 10 Tasks (B0 Inc10) $\bar{A}$ | $\mathcal{A}_B$ | CIFAR-100 20 Tasks (B0 Inc5) $\bar{A}$ | $\mathcal{A}_B$ | ObjectNet 10 Tasks (B0 Inc20) $\bar{A}$ | $\mathcal{A}_B$ | ObjectNet 20 Tasks (B0 Inc10) $\bar{A}$ | $\mathcal{A}_B$ | ImageNet-R 10 Tasks (B0 Inc20) $\bar{A}$ | $\mathcal{A}_B$ | ImageNet-R 20 Tasks (B0 Inc10) $\bar{A}$ | $\mathcal{A}_B$ | ImageNet-A 10 Tasks (B0 Inc20) $\bar{A}$ | $\mathcal{A}_B$ | VTAB 5 Tasks (B0 Inc10) $\bar{A}$ | $\mathcal{A}_B$ |
|---|---|---|---|---|---|---|---|---|---|---|---|---|---|---|---|---|
| L2P | 85.92 | 79.19 | 85.94 | 79.93 | 66.77 | 55.16 | 63.78 | 52.19 | 75.46 | 69.77 | 63.75 | 55.78 | 49.39 | 41.71 | 77.11 | 77.10 |
| DualPrompt | 89.65 | 84.89 | 87.87 | 81.15 | 64.31 | 52.99 | 59.27 | 49.33 | 73.10 | 67.18 | 66.52 | 61.77 | 53.71 | 41.67 | 83.36 | 81.23 |
| CODA-Prompt | 91.05 | 86.44 | 89.11 | 81.96 | 66.53 | 56.80 | 66.07 | 53.29 | 77.97 | 72.27 | 70.45 | 64.68 | 53.54 | 42.73 | 83.90 | 83.02 |
| SLCA | 92.67 | 89.30 | 92.49 | 88.55 | 74.12 | 63.23 | 72.55 | 61.30 | 81.17 | 77.00 | 81.85 | 76.63 | 68.66 | 58.74 | 90.94 | 90.76 |
| FeCAM | 93.23 | 89.05 | 91.86 | 87.04 | 68.68 | 57.39 | 67.46 | 54.87 | 79.02 | 72.53 | 77.84 | 71.05 | 56.04 | 46.41 | 87.38 | 82.20 |
| SSIAT | 94.35 | 91.35 | 93.52 | 90.07 | 72.45 | 62.45 | 76.56 | 65.87 | 83.20 | 78.85 | 82.30 | 75.67 | 70.83 | 62.23 | 95.19 | 90.58 |
| InfLoRA | 91.70 | 86.51 | 89.13 | 81.46 | 70.67 | 58.04 | 66.26 | 51.81 | 80.82 | 75.65 | 77.28 | 71.01 | 58.50 | 46.28 | 88.90 | 87.63 |
| EASE | 92.11 | 87.72 | 91.51 | 85.80 | 71.04 | 59.37 | 70.84 | 57.86 | 81.74 | 76.17 | 81.18 | 74.62 | 65.34 | 55.04 | 93.61 | 93.55 |
| APER-Adapter | 92.22 | 87.45 | 90.65 | 85.15 | 69.24 | 57.41 | 67.18 | 55.24 | 75.82 | 67.95 | 72.35 | 64.33 | 60.53 | 49.57 | 85.95 | 84.35 |
| COFiMA | 93.87 | 89.77 | 92.86 | 88.09 | 73.21 | 61.36 | 71.86 | 58.06 | 82.05 | 76.43 | 81.54 | 75.15 | 57.70 | 47.60 | 95.21 | 91.86 |
| SEMA | 91.60 | 86.75 | 92.23 | 87.84 | 67.95 | 54.92 | 67.95 | 54.92 | 81.39 | 77.84 | 77.84 | 69.60 | 63.83 | 52.21 | 91.99 | 90.86 |
| SD-LoRA | 92.54 | 88.01 | 90.90 | 85.18 | 70.37 | 58.54 | 67.87 | 52.78 | 82.04 | 77.34 | 80.22 | 75.26 | 64.95 | 55.96 | 89.11 | 86.54 |
| MOS | 93.83 | 90.19 | 93.30 | 89.25 | 74.75 | 65.10 | 74.69 | 63.62 | 81.74 | 76.17 | 82.96 | 77.93 | 69.13 | 59.12 | 92.62 | 92.79 |
| MoAL | 94.22 | 90.49 | 93.27 | 88.36 | 75.37 | 65.33 | 74.20 | 60.29 | 84.45 | 79.33 | 82.94 | 76.85 | **74.29** | 64.06 | 95.06 | 89.79 |
| TUNA | 94.85 | 91.71 | 94.44 | 90.74 | 76.46 | 66.32 | 75.25 | 64.33 | 84.22 | 79.42 | 82.54 | 77.45 | 72.96 | 64.38 | 94.77 | 90.33 |
| MIN | 95.12 | 92.12 | 94.31 | 91.03 | 72.56 | 61.36 | 70.92 | 61.16 | 85.18 | 79.75 | 83.69 | 78.08 | 72.89 | 64.32 | 96.47 | 92.26 |
| **CaRE (Ours)** | **95.46** | **92.46** | **95.26** | **91.97** | **77.58** | **67.55** | **77.54** | **66.54** | **85.75** | **80.53** | **84.57** | **78.80** | 73.40 | **64.78** | **96.93** | **93.80** |

$\mathcal{A}_B$, respectively. Similarly, on ImageNet-R and -A (both 50 tasks), CaRE exceeds SSIAT by 2.51% and 10.8% in $\mathcal{A}_B$. These results strongly validate that our method possesses a robust balance between plasticity and stability when handling long task sequences. Overall, to the best of our knowledge, our work is the first to successfully scale continual learning to over 300 non-overlapping tasks while maintaining clearly superior performance.

## 5.2. Short Task Sequence Evaluations

**Setup.** To evaluate the performance of our method in classical short task sequence evaluations, we conduct experiments on regular CIL settings (ranging from 5 to 20 tasks) using 5 commonly used datasets: CIFAR-100 (Krizhevsky et al., 2009), ObjectNet (Barbu et al., 2019), ImageNet-R (Hendrycks et al., 2021a), ImageNet-A (Hendrycks et al., 2021b), and VTAB (Zhai et al., 2019), following the protocols of (Zhou et al., 2025). During training, we employ the same training settings described in Section 5.1, utilizing a ViT-B/16-IN21K as the backbone and shuffling the task order using a random seed of 1993. In addition to the comparison methods in Section 5.1, we further include several competitive PTM-based CIL approaches for extended comparison: SLCA (Zhang et al., 2023), FeCAM (Goswami et al., 2023), InfLoRA (Liang & Li, 2024), COFiMA (Marouf et al., 2024), and SD-LoRA (Wu et al., 2025). Note that some of these methods cannot be seamlessly scaled to our long-sequence evaluation scenarios due to either implementation incompatibilities, training instability (e.g., training collapse when task number increases significantly), or prohibitive computational demands. Consequently, these methods are evaluated only in short-sequence settings. All baseline methods are implemented using their official implementations with the same PTM for fair comparisons.

**Results.** Table 3 shows that our method achieves leading performance on diverse datasets. For instance, on the 10-task setting of CIFAR-100, CaRE improves upon MoAL by a notable 1.97% in $\mathcal{A}_B$. On the 20-task setting of Object-

*Table 4.* Ablation study on router selection strategy.

| Method | $\bar{A}$ | $\mathcal{A}_B$ |
|---|---|---|
| Baseline | **78.54** | **68.27** |
| w/o Dynamics | 68.85 (-9.69) | 58.29 (-9.98) |
| Autoencoder | 74.29 (-4.25) | 64.35 (-3.92) |
| Cosine Head | 75.88 (-2.66) | 65.91 (-2.36) |
| Max Logits | 74.75 (-3.79) | 65.21 (-3.06) |

*Table 5.* Ablation study on MoE architecture.

| Method | $\bar{A}$ | $\mathcal{A}_B$ |
|---|---|---|
| Baseline | **78.54** | **68.27** |
| Single Router | 66.75 (-11.79) | 43.96 (-24.31) |
| w/o Gate (K=1) | 73.89 (-4.65) | 62.14 (-6.13) |
| w/o Gate (K=2) | 73.25 (-5.29) | 61.98 (-6.29) |
| w/o Gate (K=3) | 72.77 (-5.77) | 61.84 (-6.43) |

Net, CaRE surpasses SLCA by 5.24% in $\mathcal{A}_B$. Compared with SD-LoRA, CaRE improves by 3.19% and 8.82% in $\mathcal{A}_B$ on the 10-task settings of ImageNet-R and -A, respectively. Notably, even with very few tasks (i.e., 5-task setting of VTAB), CaRE maintains superior performance over all competitors, demonstrating its powerful robustness.

## 5.3. Ablation Studies

**Setup.** We conduct extensive ablation studies to validate the key design choices of CaRE. All experiments are performed on OmniBenchmark-1K with 100 tasks (B0 Inc10), using training settings consistent with Section 5.1. Due to the page limit, more results are provided in Appendix A.2.

**Analysis of dynamic router selection.** In Table 4, we systematically analyze the impact of different dynamic router selection strategies on final performance, using our final implementation as the "Baseline". At first, we remove all class perceptrons and utilize a prototype classifier to determine the task identity (Zhou et al., 2025; Sun et al., 2025b), thereby activating corresponding router networks at each layer. The resulting model is termed "w/o Dynamics", which isolates the effect of dynamically selecting the task identity per layer. This model leads to a significant per-

*Table 6.* Comparison of computational efficiency.

| Method | $P_t \downarrow$ (M) | $P_a \downarrow$ (M) | $S_t \downarrow$ (s) | $\mathcal{A}_B \uparrow$ (%) |
|---|---|---|---|---|
| L2P | 0.82 | 86.61 | 25.06 | 48.87 |
| DualPrompt | 1.02 | 86.82 | 23.05 | 49.45 |
| EASE | 1.19 | 195.77 | 1096.89 | 53.54 |
| TUNA | 16.82 | 32.58 | 820.19 | 60.04 |
| MOS | 16.14 | 32.27 | 1116.54 | 64.27 |
| MIN | 9.23 | 112.41 | 71.13 | 63.60 |
| **CaRE (Ours)** | 3.27 | 90.99 | 70.89 | 68.27 |

*Table 7.* Task-related routing recall (%) under the 301-task evaluation protocol. R@2 denotes Top-2 router recall, and E@3 denotes Top-3 expert recall.

| Layer | T=10 | | T=100 | | T=301 | |
|---|---|---|---|---|---|---|
| | R@2 | E@3 | R@2 | E@3 | R@2 | E@3 |
| 3 | 55.0 | 83.5 | 33.5 | 63.2 | 27.4 | 58.4 |
| 6 | 65.2 | 88.7 | 46.8 | 71.6 | 40.3 | 66.1 |
| 9 | 80.5 | 93.1 | 71.3 | 81.5 | 66.8 | 76.2 |
| 12 | 92.3 | 96.7 | 85.1 | 90.2 | 80.6 | 85.8 |

formance drop, demonstrating the importance of dynamic modeling. Furthermore, we replace the entropy-based class perceptrons with several alternatives: (1) An "Autoencoder" trained with an MSE loss to memorize the task distribution, selecting router networks at inference based on the minimal reconstruction error; (2) A method inspired by the prototype classifier (Zhou et al., 2025), where each class perceptron maintains a set of prototypes per task and selects router networks via cosine similarity at inference, termed "Cosine Head"; (3) A naive strategy where, for each task, the maximum value from its class perceptron's softmax logits is taken, and these per-task maximum values are then sorted to determine the layer-wise task identity. All these alternatives result in notable performance degradation, validating the effectiveness of using entropy to dynamically select layer-wise task identity, as discussed in Section 3.4.

**Effect of MoE architecture.** As shown in Table 5, we conduct a comprehensive ablation study to validate the effect of the MoE architecture in CaRE. First, instead of instantiating a new task-specific router for each arriving task, we employ a channel-expansion router (Wang et al., 2025a) that enlarges the channel dimension of a single shared router for every new task, while removing all class perceptrons. This variant is denoted as "Single Router". Results show a dramatic performance drop, indicating that a single router cannot adequately accommodate knowledge from an increasing number of tasks. Second, we eliminate the dynamic gating mechanism of the router in MoE, i.e., we directly select **K** task-specific experts via the class perceptrons and sum their outputs. This model is referred to as "w/o Gate". It can be observed that performance declines noticeably without the MoE gating mechanism. Increasing the number of task-specific experts (**K** from 1 to 3) without gating further degrades performance, since simple unweighted summation may yield task-incompatible representations. Overall, these results collectively demonstrate the importance of the MoE architecture in our method.

**5.4. Analytical Experiments**

**Computational efficiency.** Under the 100-task setting, we analyze computational efficiency from three perspectives: the average number of trainable parameters per task ($P_t$), the additional parameters appended to the PTM after learn-

ing all tasks ($P_a$), and the average inference latency ($S_t$). As shown in Table 6, CaRE achieves an excellent trade-off between performance and efficiency. Specifically, compared with MOS, CaRE improves $\mathcal{A}_B$ by 4% while using approximately 80% fewer average trainable parameters and 95% lower inference latency. Compared with MIN, CaRE improves $\mathcal{A}_B$ by 4.67% with comparable inference latency and fewer trainable parameters.

**Router and expert activation patterns.** We further analyze task-related routing by measuring the recall of activated routers and experts across layers and task scales. For routers, we report Top-2 recall: a prediction is counted as correct if either of the two activated routers matches the ground-truth task. Similarly, for experts, we report Top-3 recall. We conduct this analysis under the 301-task evaluation protocol, and report the results after learning 10, 100, and all 301 tasks. As shown in Table 7, recall consistently increases from shallow to deep layers for both routers and experts. This trend suggests that deeper layers produce increasingly task-specific representations, thereby enabling more reliable routing decisions. As the task pool grows from 10 to 301, the candidate space also expands substantially, making the routing problem increasingly challenging. Nevertheless, the final layer still achieves 80.6% router recall and 85.8% expert recall at T=301, confirming that CaRE maintains effective task-related routing even at large task scales. More analysis and discussions are provided in Section A.3.

## 6. Conclusion

This paper proposes CaRE, a novel PTM-based continual learner featuring an efficient BR-MoE. The core of BR-MoE is a simple yet effective bi-level routing mechanism that enables dynamic knowledge retrieval and aggregation at each hidden layer of a continual learner. Additionally, we introduce OmniBenchmark-1K, a challenging long-sequence CIL benchmark designed to facilitate scalable evaluation with hundreds of tasks. Extensive experiments across diverse datasets demonstrate the effectiveness of our method, particularly in long-sequence scenarios (100 to 301 tasks). We hope this work can encourage the development of more scalable and practical continual learning systems, which are capable of operating effectively in real-world environments with potentially unbounded task sequences.

## Acknowledgements

This work was supported in part by Hong Kong Research Grants Council under NSFC/RGC Collaborative Research Scheme (Grant CRS_HKU703/24) and Shenzhen Loop Area Institute (Grant FPF10120250002).

## Impact Statement

This work proposes a generic continual learning framework. All experiments are conducted on publicly available datasets and utilize open-source pre-trained models, ensuring that no private or sensitive data are involved. Furthermore, the research is domain-agnostic and does not raise ethical and societal concerns, as it is not designed for potentially harmful applications such as surveillance or misinformation dissemination.

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

# A. Appendix

## A.1. More Experimental Comparisons

**Impact of different task orders.** To evaluate the robustness of different PTM-based CIL methods in long-sequence evaluations, we conduct experiments on OmniBenchmark-1K with different task orders. Specifically, based on the 100-task results in Table 1, we additionally generate 3 distinct task sequences using random seeds of $\{1990, 1996, 1999\}$, and report the mean and standard deviation (std) of performance across these runs. As listed in Table 8, CIL methods such as MOS and TUNA exhibit relatively high std in $\mathcal{A}_B$ (0.88 and 1, respectively), indicating their sensitivity to task ordering in long-sequence evaluations. In contrast, our CaRE achieves the lowest std (0.16 in $\mathcal{A}_B$), demonstrating more powerful robustness compared to strong baselines.

**Impact of different pre-trained weights.** In addition to the ViT-B/16-IN21K backbone, we evaluate all methods using the ViT-B/16-IN1K model under the 100-task setting (B0 Inc10) with OmniBenchmark-1K. As shown in Table 9, our CaRE consistently outperforms all baselines by a clear margin. For instance, it surpasses the strong baseline MIN by 6.99% in $\mathcal{A}_B$, further demonstrating the robustness of our method across different PTMs.

## A.2. More Ablation Studies

Building on the training settings described in Section 5.3, we present additional ablation studies to comprehensively analyze the contribution of each component in our proposed method.

**Effect of the local decision scope.** To assess the efficacy of layer-wise local decisions, we vary the extent to which router selections are shared across successive BR-MoE layers. Specifically, we introduce a scope hyperparameter: Scope = 1 allows each layer to select routers independently (our baseline). Scope = 2 reuses the current layer's activated router indices for the next layer. Similarly, Scope = 3 propagates the current layer's selections to

*Table 8.* Performance on 100 tasks (B0 Inc10) with different task orders.

| Method | $\bar{\mathcal{A}}$ | $\mathcal{A}_B$ |
|---|---|---|
| L2P | $61.14 \pm 1.21$ | $47.94 \pm 0.68$ |
| DualPrompt | $62.54 \pm 0.84$ | $48.92 \pm 0.39$ |
| CODA-Prompt | $64.94 \pm 1.02$ | $51.32 \pm 0.29$ |
| EASE | $65.78 \pm 1.05$ | $53.17 \pm 0.35$ |
| SSIAT | $75.41 \pm 0.89$ | $63.05 \pm 0.43$ |
| APER-Adapter | $73.66 \pm 0.93$ | $62.48 \pm 0.16$ |
| SEMA | $56.97 \pm 0.32$ | $32.23 \pm 1.50$ |
| MoAL | $63.91 \pm 0.98$ | $41.33 \pm 0.59$ |
| TUNA | $71.84 \pm 0.61$ | $58.79 \pm 1.00$ |
| MOS | $78.17 \pm 1.03$ | $65.58 \pm 0.88$ |
| MIN | $76.12 \pm 0.84$ | $63.75 \pm 0.21$ |
| **CaRE (Ours)** | $\mathbf{78.67 \pm 0.58}$ | $\mathbf{68.06 \pm 0.16}$ |

*Table 9.* Performance of the ViT-B/16-IN1K model under the 100-task setting (B0 Inc10).

| Method | $\bar{\mathcal{A}}$ | $\mathcal{A}_B$ |
|---|---|---|
| L2P | 57.92 | 45.76 |
| DualPrompt | 58.89 | 47.18 |
| CODA-Prompt | 54.40 | 49.78 |
| EASE | 65.15 | 53.52 |
| SSIAT | 74.70 | 63.65 |
| APER-Adapter | 71.37 | 60.17 |
| SEMA | 50.69 | 27.46 |
| MoAL | 65.45 | 43.27 |
| TUNA | 73.16 | 61.72 |
| MOS | 76.47 | 64.17 |
| MIN | 73.47 | 60.65 |
| **CaRE (Ours)** | **77.76** | **67.64** |

*Table 10.* Performance under different local decision scopes.

| Scope | $\bar{\mathcal{A}}$ | $\mathcal{A}_B$ |
|---|---|---|
| 1 (Baseline) | **78.54** | **68.27** |
| 2 | $77.24_{(-1.30)}$ | $67.15_{(-1.12)}$ |
| 3 | $76.15_{(-2.39)}$ | $66.25_{(-2.02)}$ |

the next two layers. As shown in Table 10, performance degrades steadily as the Scope increases, indicating that each layer benefits from a customized knowledge-retrieval pattern and supporting our insights in Section 1.

**Effect of the number of activated router networks.** We evaluate the impact of the number of router networks activated during the forward pass in each BR-MoE layer, corresponding to the hyper-parameter $\mathbf{M}$ introduced in Section 3.3. In addition to the default $\mathbf{M}=2$ (baseline), we test $\mathbf{M}=1, 3$, and 4, respectively. We also construct an alternative model which only activates a single router network ($\mathbf{M}=1$) but activates twice as many adapters (experts). As shown in Table 11, increasing $\mathbf{M}$ from 1 to 2 yields a notable improvement of 1.37% in $\bar{\mathcal{A}}$ and 1.20% in $\mathcal{A}_B$. This confirms that reusing historical routers effectively enhances long-sequence evaluations, aligning with the discussion in Section 3.4. However, further increasing $\mathbf{M}$ to 3 and 4 leads to a gradual performance decline, suggesting that more router networks may introduce irrelevant information. Furthermore, the $\mathbf{M}=1$ variant with twice as many experts does not outperform the default $\mathbf{M}=2$ setup. Although the total number of activated experts is similar, the latter leverages previously trained router networks to selectively compose features from historical experts, thereby better integrating cross-task knowledge. These results further validate the effectiveness of our bi-level routing mechanism.

**Effect of the number of activated experts.** We investigate the effect of the number of activated experts (adapters) in BR-MoE, controlled by the hyperparameter $\mathbf{K}$ introduced in Section 3.3. Our default setting uses $\mathbf{K}=3$, where each router activates its Top-3 experts. We compare this with

*Table 11.* Ablation study on the number of activated router networks.

| Method | $\bar{\mathcal{A}}$ | $\mathcal{A}_B$ |
|---|---|---|
| **M = 2 (Baseline)** | **78.54** | **68.27** |
| **M = 4** | 78.00$_{(-0.54)}$ | 67.99$_{(-0.28)}$ |
| **M = 3** | 78.26$_{(-0.28)}$ | 68.06$_{(-0.21)}$ |
| **M = 1** | 77.17$_{(-1.37)}$ | 67.07$_{(-1.20)}$ |
| **M = 1 (More Experts)** | 77.25$_{(-1.29)}$ | 67.05$_{(-1.22)}$ |

*Table 12.* Ablation study on the number of activated adapters.

| Method | $\bar{\mathcal{A}}$ | $\mathcal{A}_B$ |
|---|---|---|
| **K = 3 (Baseline)** | **78.54** | **68.27** |
| **K = 1** | 74.55$_{(-3.99)}$ | 64.12$_{(-4.15)}$ |
| **K = 2** | 78.28$_{(-0.26)}$ | 68.14$_{(-0.13)}$ |
| **K = 6** | 78.48$_{(-0.06)}$ | 68.25$_{(-0.02)}$ |
| **K = 18** | 78.45$_{(-0.09)}$ | 68.17$_{(-0.10)}$ |

alternative settings $\mathbf{K} \in \{1, 2, 6, 18\}$. Note that since each router network can only access the experts corresponding to the current task and previous tasks, for larger $\mathbf{K}$, early tasks cannot activate as many experts during training due to the limited number of existing experts at that time. As shown in Table 12, performance improves significantly when increasing $\mathbf{K}$ from 1 to 2, since activating only a single expert loses the characteristic of the MoE architecture and cannot effectively utilize knowledge from different experts. However, performance saturates beyond $\mathbf{K} = 3$, with no gains observed for $\mathbf{K} = \{6, 18\}$. This phenomenon diverges from observations in MoE-based Large Language Models (LLMs), where activating more experts typically improves performance (Wu et al., 2024; Jie et al., 2025). The difference stems from the distinct objective of continual learning: rather than seeking generic knowledge aggregation, the continual learner should perform precise knowledge retrieval from a growing set of historical tasks. Hence, activating too many experts inevitably introduces features from unrelated tasks, which act as noise and dilute the discriminative power required for accurate predictions. The saturation at $\mathbf{K} = 3$ indicates that our bi-level routing mechanism successfully identifies a small set of highly relevant experts, while adding further experts provides negligible complementary information, increasing the risk of interference. This result validates that selective and precise knowledge retrieval, rather than merely increasing expert participation, is crucial for maintaining robustness in long-sequence evaluations.

**Analysis of the class perceptron.** In Table 13, we analyze the effect of the additional loss introduced for the class perceptron, as mentioned in equation 8. Specifically, we first remove $\mathcal{L}_{\text{KL}}^{\ell}$ to examine the effectiveness of directly using the softmax logits generated from high-level features as the supervision. The results show a slight performance drop. Furthermore, we evaluate different values of the scaling

*Table 13.* Ablation study on the auxiliary loss of the class perceptron.

| Method | $\bar{\mathcal{A}}$ | $\mathcal{A}_B$ |
|---|---|---|
| Baseline | **78.54** | **68.27** |
| w/o $\mathcal{L}_{\text{KL}}^{\ell}$ | 78.13$_{(-0.41)}$ | 68.03$_{(-0.24)}$ |
| $\lambda = 0.5$ | 78.36$_{(-0.18)}$ | 68.24$_{(-0.03)}$ |
| $\lambda = 1.5$ | 78.34$_{(-0.20)}$ | 68.24$_{(-0.03)}$ |
| $\lambda = 2.0$ | 78.23$_{(-0.31)}$ | 68.20$_{(-0.07)}$ |

*Table 14.* Ablation study on adapter configuration.

*(a)* Number of channels in a regular expert.

| Channel | $\bar{\mathcal{A}}$ | $\mathcal{A}_B$ |
|---|---|---|
| 8 | 78.44 | 68.14 |
| 16 | **78.54** | **68.27** |
| 32 | 78.48 | 68.14 |
| 64 | 78.52 | 68.21 |

*(b)* Number of channels in the shared expert.

| Channel | $\bar{\mathcal{A}}$ | $\mathcal{A}_B$ |
|---|---|---|
| None | 78.32 | 67.84 |
| 32 | 78.51 | 68.13 |
| 64 | **78.54** | **68.27** |
| 96 | 78.31 | 68.04 |
| 128 | 78.36 | 67.90 |

*(c)* EMA decay $\mu$ in the shared expert.

| $\mu$ | $\bar{\mathcal{A}}$ | $\mathcal{A}_B$ |
|---|---|---|
| 0.9 | 78.19 | 68.11 |
| 0.99 | 78.50 | 68.24 |
| 0.999 | **78.54** | **68.27** |
| 0.9999 | 78.29 | 68.09 |

factor $\lambda$ (in equation 9), including 0.5, 1.5, and 2.0. The results demonstrate minimal performance variation across these settings, suggesting that although the class perceptron introduces auxiliary supervision, the overall performance is not sensitive to the weighting of this loss.

**Effect of different configurations of adapter.** We conduct a series of experiments to analyze key design choices in our adapter (expert) modules. First, we vary the bottleneck channel size of each task-specific expert among $\{8, 16, 32, 64\}$. As shown in Table 14 (a), a channel size of 16 yields the highest $\bar{\mathcal{A}}$ and $\mathcal{A}_B$, indicating that 16 channels are sufficient to encapsulate task-specific information. Based on this, we further examine the configuration of the shared expert. Specifically, we compare removing the shared expert against varying its bottleneck channel size in $\{32, 64, 96, 128\}$. Results in Table 14 (b) show that including a shared expert with 64 channels provides a moderate improvement of 0.43% in $\mathcal{A}_B$. Notably, the shared expert remains a single and unified module throughout the entire CIL process for each BR-MoE module. Finally, we analyze the EMA decay coefficient

*Table 15.* Comparison of different module placement strategies.

| Placement | $\bar{\mathcal{A}}$ | $\mathcal{A}_B$ |
|---|---|---|
| After Adapter | **78.54** | **68.27** |
| Before Adapter | $77.18_{(-1.36)}$ | $67.74_{(-0.53)}$ |

$\mu$ for updating the shared expert (equation 6). A larger $\mu$ causes the shared expert to evolve more slowly, retaining more knowledge from earlier tasks. As reported in Table 14 (c), $\mu = 0.999$ achieves the best trade-off, allowing the shared expert to stably accumulate cross-task knowledge.

**Analysis of module placement.** The relative placement of class perceptrons and router networks with respect to the task-specific adapters influences how routing decisions are made. There are two variants: (1) Before Adapter: both the class perceptron and the router network of each task receive the original input feature directly. (2) After Adapter: the input is first transformed by its task-specific adapter, and the resulting adapted feature is then fed to the corresponding class perceptron and router network. The rest of the forward process remains identical to Section 3.3. Results in Table 15 show that the "After Adapter" design yields consistently better performance. This suggests that routing based on task-adapted features provides more discriminative and stable signals for both context encoding and expert selection, thereby improving overall routing reliability and final prediction accuracy. Note that Figure 2 illustrates the "Before Adapter" variant for clearer visualization of the overall workflow of our method.

### A.3. More Analytical Experiments

**Visualization analysis of bi-level routing.** We conduct a visualization analysis to more intuitively understand our bi-level routing mechanism. Specifically, using the CaRE model trained under the 100-task setting, we visualize the routing process in the final layer of the model. As shown in Figure 3 (a), when a Corgi image from Task 96 is processed, two router networks are activated through entropy-based selection: the primary router corresponds to Task 96 (correctly identifying the current task), while the secondary router corresponds to Task 53. These routers then activate distinct sets of adapter experts. Router 96 activates experts $\{96, 16, 37\}$, where expert 96 is task-specific, while experts 16 and 37 include animal-related knowledge, particularly expert 16, which has learned dog-related features and thus receives higher gating weights. Router 53 activates experts $\{53, 19, 1\}$, all of which contain animal knowledge. Notably, expert 53 includes the Husky class that shares visual similarities with Corgis, thereby providing complementary information for the final prediction. We also visualize class activation map via the Grad-CAM technique (Selvaraju et al., 2020). The output from router 96 empha-

sizes facial characteristics (discriminative representations), whereas router 53 captures shared details such as ear shape and texture (complementary representations). The resulting feature map aggregates both discriminative and complementary cues, yielding a more accurate representation. Similarly, as shown in Figure 3 (b), the two router networks produce feature maps that focus on different cues, enabling the final output feature map to more accurately capture the object region.

**Visualization of router and expert activation patterns.** To gain deeper insight into the dynamic router selection and dynamic expert routing mechanisms of the proposed BR-MoE across the different network layers, we have conducted a comprehensive visualization analysis of its activation patterns. Using the model trained on the 100-task evaluation protocol, we perform inference on the entire validation set, covering statistics across all 100 tasks. For each task, we compute two metrics: the activation frequency of each router network and the utilization rate of each adapter expert (calculated as the multiplication of its activation frequency and the corresponding gating scores). We visualize these patterns concerning four layers $\{3, 6, 9, 12\}$, which refer to the progression from shallow to deep representations.

As shown in Figure 4, the activation patterns exhibit a clear hierarchical structure. In the earlier layers (e.g., layers 3 and 6), a small subset of router networks (a) and experts (b) is activated with high frequency across a broad range of tasks. These findings align with the fact that early layers capture low-level visual commonalities like edges and textures (Lou et al., 2025b; Lou & Yu, 2025; Fu et al., 2025). By acting as robust feature extractors for shared elements, these frequently activated components facilitate a relatively generalized low-level representation in early layers.

In contrast, deeper layers (e.g., layers 9 and 12) exhibit significantly sparser and more task-specific activation patterns. In these layers, the utilization of both router networks (a) and experts (b) becomes increasingly concentrated, i.e., the router networks and experts corresponding to the ground-truth task identity are activated with substantially higher intensity. This pronounced shift indicates that high-level semantic representations are highly specialized, necessitating the extraction of more discriminative features to ensure accurate classification. Note that the relatively weak activation intensity of experts in later tasks does not stem from underutilization or expert collapse. This phenomenon occurs because experts from earlier tasks can be continually revisited and reactivated by subsequent tasks for knowledge reuse, thereby accumulating higher gating scores over tasks.

It is worth mentioning that the visualization reveals two phenomena that empirically support the core insights discussed in Section 1. First, in each layer, the model not only activates task-specific components but also incorporates knowledge

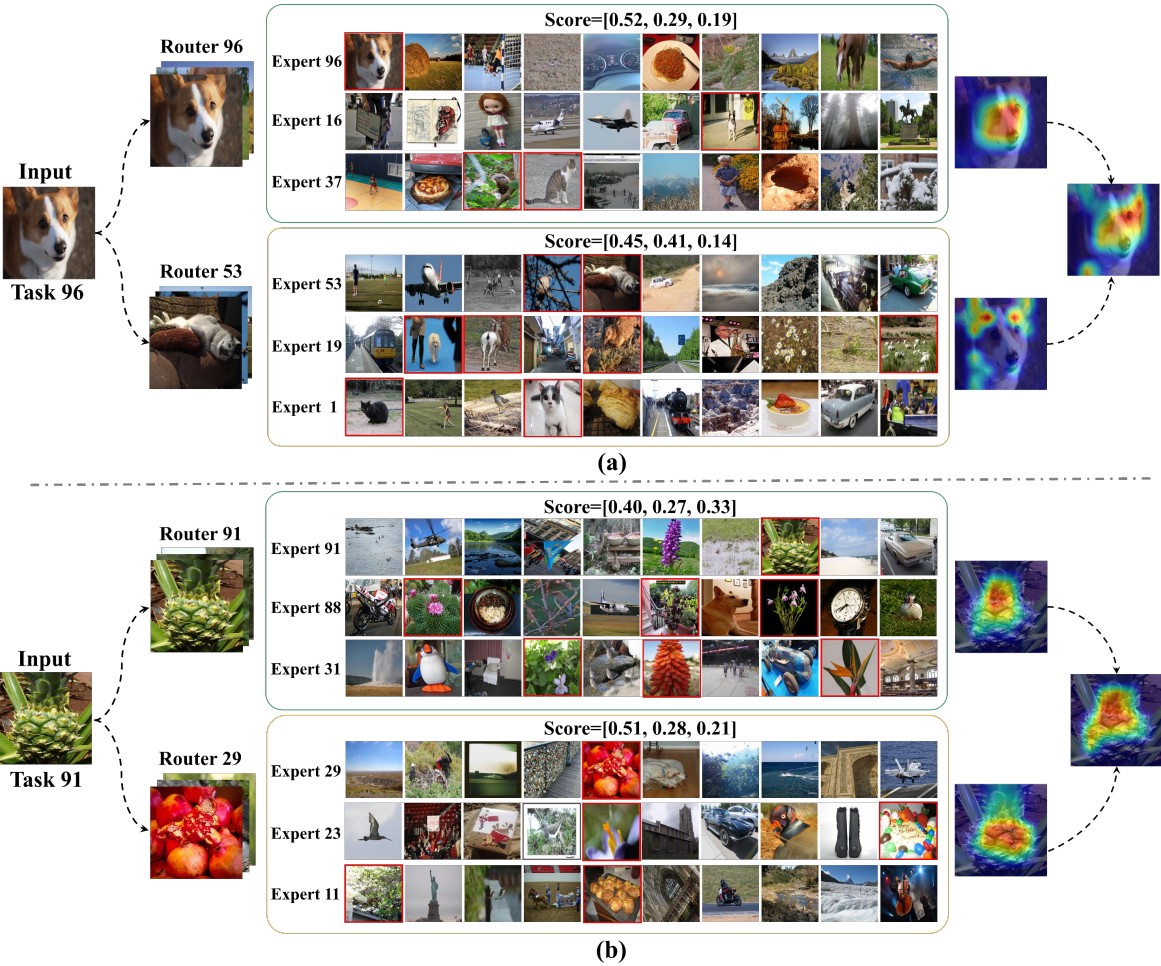

*Figure 3.* Visualization of our bi-level routing mechanism.

from different tasks, resulting in features that are both discriminative (derived from task-specific routers and experts) and comprehensive (derived from complementary routers and experts). Our ablation study in Section A.2 empirically demonstrates the importance of this property, by showing that restricting the model to a single router or expert results in notable performance degradation.

Second, the decision patterns diverge markedly across different layers. This variation primarily arises from the fact that different layers possess distinct levels of semantic abstraction, whereas our method allows each layer to retrieve the most pertinent knowledge according to its specific-level of abstraction. This aligns with experimental results aforementioned, where our method demonstrates a significant advantage over strong adapter-based CIL baselines that bind task-specific adapters at every layer (e.g., MOS (Sun et al., 2025b) and TUNA (Wang et al., 2025b)). Furthermore, the ablation studies in Sections 5.3 and A.2 confirm that removing layer-wise dynamics or expanding the local decision scope negatively impacts performance.

We also observe an interesting phenomenon: even though parameters learned in subsequent tasks are unavailable during the training of early tasks, knowledge from these later tasks is still incorporated by the model on earlier tasks during inference. This suggests that the test-time flexibility of our bi-level routing mechanism, which enables the active integration of knowledge from different tasks at test-time, regardless of task order. We believe that optimizing performance at test-time may be a promising direction for future continual learning research.

Overall, these results provide an intuitive validation of the core mechanisms driving CaRE: 1) discriminative and comprehensive representation learning; 2) dynamic layer-wise local decision-making. Both properties are key contributors to the clearly superior performance of our CaRE in both long- and short-sequence CIL evaluations over existing methods.

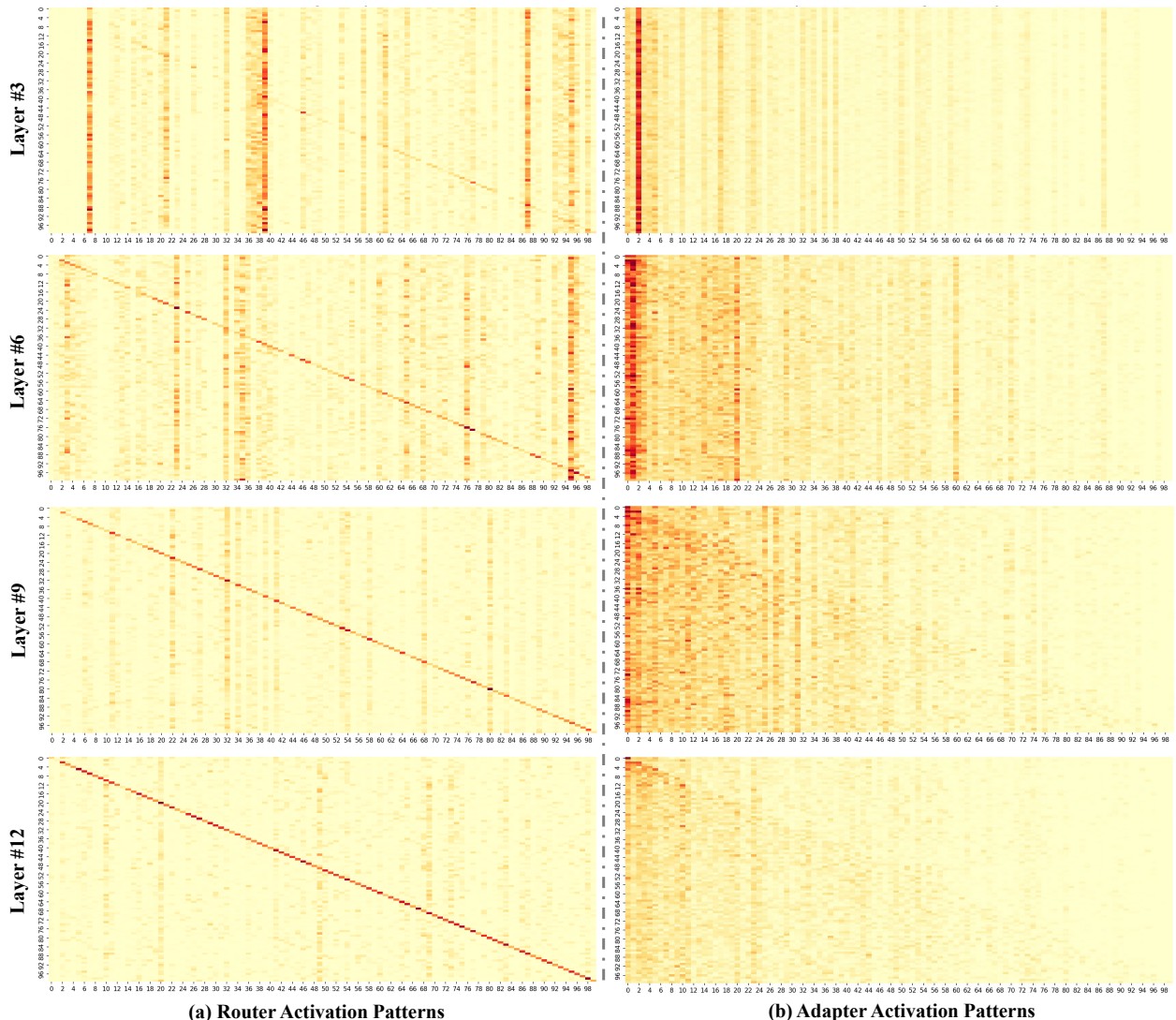

*Figure 4.* Visualization of bi-level router and expert activation patterns. **(a)** Dynamic router selection (first level) patterns using class perceptrons. **(b)** Dynamic adapter/expert selection patterns (second level). Rows and columns of each heatmap correspond to router/expert indices and task indices, respectively.

### A.4. Limitations

To the best of our knowledge, this work presents the first continual learner scaled to more than 300 non-overlapping tasks. However, similar to existing popular PEFT-based approaches, our method still relies on appending new efficient modules for each task, leading to model complexity that grows linearly with the number of tasks. While investigating even longer or potentially unbounded task sequences remains highly relevant for real-world applications, such extensive evaluation is beyond the scope of this study due to the lack of computational resources, and we plan to further improve computational efficiency in the long-sequence evaluation protocol. Nevertheless, given the simplicity and strong performance of our method, we hope that both our approach and the introduced dataset can serve as a foundation for future research on continual learning under extremely long task sequences. Promising directions include further architectural simplification without sacrificing performance, as well as extending the long-sequence evaluation protocol to vision-language models (Radford et al., 2021; Yang et al., 2025; Shi et al., 2026).

