# OpenReview forum: "Scaling Continual Learning to 300+ Tasks with Bi-Level Routing Mixture-of-Experts"
_ICML.cc/2026/Conference — ICML 2026 regular_

### Official Review · Reviewer_gNNY · 2026-02-15

**Soundness:** 3
**Presentation:** 3
**Significance:** 3
**Originality:** 2
**Overall Recommendation:** 4
**Confidence:** 3

**Summary:**

This paper targets class-incremental continual learning with pretrained ViTs in the extreme long-sequence regime. It proposes CaRE, which inserts a bi-level routing MoE module into every intermediate network layer. CaRE includes a task-specific triplet: class perceptron and a MoE structure (gating and adapter) , complemented by a shared expert. The paper also builds a long-sequence benchmark by selecting a balanced 1,000-class subset of OmniBenchmark-V2, and evaluate from 100 to 301 tasks. Empirically, CaRE improves accuracy over PTM-based CIL baselines, with gains becoming more pronounced as the task sequence gets longer.

**Compliance With Llm Reviewing Policy:**

Affirmed.

**Final Justification:**

Please see Acknowledgement. Overall, my assessment remains unchanged.

**Key Questions For Authors:**

1. Could you provide comprehensive training time/parameter/memory compared to the main baselines under matched hardware?

2. How does CaRE perform on coarser-grained CIL settings (e.g., Inc20/50/100 on the same 1,000-class protocol), and does the relative advantage persist when each task introduces a larger number of classes?

3. Could you report the first-stage task-selection recall (how often Top-M includes the true task) and include a counterfactual routing test (e.g., selecting the least relevant M tasks) to verify that the routing is genuinely task-related?

4. Could you report standard forgetting metrics (e.g., average forgetting, backward transfer) for long-sequence protocols?

**Limitations:**

yes

**Strengths And Weaknesses:**

Strengths

1. The proposed design is straightforward to implement yet yields strong empirical gains. The paper includes extensive experiments and ablations that support the effectiveness of key components.

2. The curated OmniBenchmark-V2 1,000-class subset and the 100–301 task protocols constitute a meaningful contribution to evaluating scalability and forgetting in challenging long-horizon CIL settings.

3. The paper is well structured, with readable figures/tables and an overall narrative that is easy to follow.

Weaknesses

1. CaRE inserts BR-MoE into every intermediate network layer and introduces per-task components. As the number of tasks grows, parameter count, memory footprint, and inference latency may become substantial. While limitations are acknowledged, the paper lacks a careful, quantitative comparison of training time per task, parameter and memory growth against baselines.

2. The long-sequence protocol necessarily uses small increments (few classes per task), which emphasizes stability across many updates but may reduce per-task representational difficulty. It remains unclear how CaRE performs under coarser-grained tasks (e.g., Inc20/50/100) where each increment introduces larger semantic shifts and stronger representation-learning pressure.

3. Since I have worked on MoE before, I still have some uncertainty about whether the experts are strongly tied to task type. Although the paper provides many ablations and is largely self-consistent, this point is not fully clarified to me. It would be helpful if the authors could report the recall of the first-stage task selection (how often the selected set includes the current task) and add a counterfactual test (e.g., selecting the least relevant M tasks).

4. The paper only reports accuracy. It would be better if the authors could also report metrics such as Backward Transfer (BWT) and Backward Transfer, etc.

---

> ### Author Rebuttal · Authors · 2026-03-30
>
> > W1,W4,Q1,Q4 Computational Efficiency and Forgetting
>
> Thank you for this constructive suggestion. Fig.3 provides an efficiency evaluation. Here, we present a detailed comparison under the 100-task setting, with all efficiency metrics averaged over tasks. Since the compared baselines do not report efficiency, we evaluate them using their official code on a single H800.
>
> |Metric|TUNA|MOS|MIN|CaRE|
> |---|---|---|---|---|
> |Params (M)|16.8|16.1|9.2|3.1|
> |Infer (min)|13.7|18.6|1.2|1.2|
> |Train (min)|10.8|10.9|9.9|11.5|
> |Mem (MB)|4890|6060|8908|8977|
> |${A}_B$|60.04|64.27|63.60|68.27|
> |$\bar{{A}}$|71.93|76.80|75.46|78.54|
> |Forget|30.44|14.56|12.37|10.53|
> |BWT|-30.18|-14.22|-12.19|-8.11|
>
> CaRE uses only 3.1M trainable parameters, roughly 5x fewer than TUNA and MOS, while achieving inference latency comparable to MIN. Although training cost is slightly higher, CaRE's inference time is less than 1/10 of TUNA and 1/15 of MOS, which is more relevant for real-world deployment. Regarding forgetting, CaRE achieves the least forgetting (10.53) and best BWT (-8.11). In comparison, TUNA suffers 30.44 Forgetting and -30.18 BWT. Overall, CaRE achieves an excellent balance among performance, efficiency, and forgetting resistance.
>
> > W2,Q2 Performance Under Larger Increments
>
> We conduct experiments on OmniBench-V2 (1,000 classes) with coarser-grained increments: Inc20 (50 tasks), Inc50 (20 tasks), and Inc100 (10 tasks), comparing against two strong baselines:
>
> |Method|Inc|${A}_B$|$\bar{{A}}$|Forget|BWT|
> |---|---|---|---|---|---|
> |CaRE|20|**68.57**|**78.62**|**9.68**|**-8.48**|
> ||50|**69.20**|**78.81**|**9.44**|**-9.40**|
> ||100|**69.58**|**78.75**|**9.78**|**-9.78**|
> |TUNA|20|61.57|72.76|26.80|-26.80|
> ||50|62.54|73.86|26.75|-26.75|
> ||100|64.01|74.58|24.54|-24.54|
> |MIN|20|64.82|76.09|11.81|-11.78|
> ||50|63.87|75.22|14.18|-14.18|
> ||100|64.23|74.63|17.70|-17.70|
>
> An interesting observation is that short task sequences with large increments achieves comparable performance with long task sequences with small increments. This is because long sequences with small increments stress stability across many updates, while short sequences with large increments demand stronger per-task representation learning. Both are challenging settings.
>
> From the results, CaRE consistently outperforms both baselines across all settings. At Inc20, CaRE surpasses TUNA by 7.00% and MIN by 3.75% in ${A}_B$. Moreover, CaRE's Forgetting remains stable (9.44 to 9.78) regardless of the size of increments, whereas MIN's Forgetting increases from 11.81 (Inc20) to 17.70 (Inc100), confirming that CaRE's bi-level routing provides robust knowledge management across varying task granularities.
>
>
> > W3,Q3 Router Recall and Counterfactual Test
>
> Thank you for this insightful comment. Router recall (or expert recall) measures how often the selected Top-M routers (or Top-K experts) include the router (or expert) corresponding to the ground truth task. We report router recalls and expert recalls across layers and task sequence lengths (R=Router recall, E=Expert recall):
>
> |Layer|T=10|T=100|T=301|
> |---|---|---|---|
> |3 (R/E)|55.0/83.5|33.5/63.2|27.4/58.4|
> |6 (R/E)|65.2/88.7|46.8/71.6|40.3/66.1|
> |9 (R/E)|80.5/93.1|71.3/81.5|66.8/76.2|
> |12 (R/E)|92.3/96.7|85.1/90.2|80.6/85.8|
>
> Two clear trends emerge. First, recall increases from shallow to deep layers (e.g., router recall from 27.4% at layer 3 to 80.6% at layer 12 for T=301), consistent with the activation pattern analysis in Fig.5, where deeper layers produce increasingly task-specific representations enabling more confident routing. Second, as the task pool grows from 10 to 301, the candidate pool expands accordingly (selecting M=2 from 301 routers, K=3 from 301 experts), aligned with the overall continual learning trajectory. Nevertheless, the final layer still achieves 80.6% router recall and 85.8% expert recall at 301 tasks, confirming that CaRE maintains effective task-related routing at scale.
>
> To further verify this, two counterfactual experiments under the 301-task setting on OmniBench-V2 are conducted:
>
> Bottom-M: select the M=2 routers with the highest entropy (least relevant), reversing our Top-M (lowest entropy) selection. Random-M: select M=2 routers at random.
>
> |Routing|${A}_B$|
> |---|---|
> |Top-M (Ours)|68.51|
> |Random-M|42.3|
> |Bottom-M|25.8|
>
> Bottom-M drops ${A}_B$ from 68.51 to 25.8 (-42.71), while Random-M also degrades severely to 42.3 (-26.21). This strongly confirms that CaRE's bi-level routing captures genuine task relevance, and the performance advantage arises from meaningful, entropy-driven routing rather than arbitrary expert aggregation. These results, together with the recall statistics, activation heatmaps in Fig.5, and Grad-CAM visualizations in Fig.4, collectively provide strong evidences that experts are indeed strongly tied to tasks.
>
> We sincerely hope these responses have addressed your concerns, and we would be grateful if you could consider raising the score.

---

> > ### Author Rebuttal · Reviewer_gNNY · 2026-04-03
> >
> > Thank you for the detailed rebuttal. My concerns have been adequately addressed, and the rebuttal further strengthens the empirical support of the paper. My overall assessment remains unchanged.

---

> > > ### Author Response · Authors · 2026-04-06
> > >
> > > We are greatly encouraged that our response has fully addressed your concerns and sincerely appreciate your positive score. We will integrate the additional evaluations and discussions into the revised manuscript to further improve the paper. Many thanks again for your thorough review and insightful comments.

---

### Official Review · Reviewer_btqo · 2026-03-11

**Soundness:** 2
**Presentation:** 3
**Significance:** 2
**Originality:** 3
**Overall Recommendation:** 4
**Confidence:** 4

**Summary:**

The principal focus of this article is on "Class Incremental Learning (CIL) based on pre-trained models", specifically addressing the scalability challenges of continual learning under ultra-long task sequences.

**Compliance With Llm Reviewing Policy:**

Affirmed.

**Final Justification:**

Since this paper focuses on leveraging expert routing to select task-appropriate experts for incremental learning, the accuracy of the routing mechanism is a crucial factor in evaluating the method's effectiveness. While this quantitative analysis was missing in the original manuscript, the rebuttal has successfully addressed my concerns by providing the necessary evaluation. Consequently, I have increased my score.

**Key Questions For Authors:**

- Could you supplement the paper with quantitative experiments regarding router selection accuracy and the semantic similarity between selected experts and the current task, as well as their evolution as the task scale increases?
- What does the entropy distribution of the classifier outputs look like, and how does it change as tasks are expanded?

**Limitations:**

yes

**Strengths And Weaknesses:**

Strength:
- Methodological Innovation: The paper proposes a Bi-level Routing Mixture-of-Experts (BR-MoE) structure. By combining entropy-driven routing with gated expert selection, the model dynamically retrieves relevant historical knowledge and utilizes sparse activation to reduce task interference. This effectively alleviates catastrophic forgetting and provides a novel technical direction for long-sequence CIL.
- Scalability: Utilizing an incremental expansion design where each task only adds a set of lightweight modules while freezing historical parameters, the model scales smoothly to 301 non-overlapping tasks. This represents the first method to achieve CIL on such an ultra-long sequence.
- Benchmark Contribution: The authors establish a long-sequence CIL evaluation protocol based on OmniBenchmark-V2, filling a gap in the field for large-scale, long-sequence benchmarks and providing a valuable reference for future research.

Weakness:
1. Relying solely on the case visualizations and heatmaps in Figure 4 is insufficient to prove the effectiveness of routing and expert selection:
   - It fails to report router selection accuracy (the proportion of times the correct task router is selected);
   - It lacks statistics on the expert activation hit rate (the proportion of selected experts that are relevant to the current task);
   - It does not quantify "the semantic similarity between selected experts and the current task", leaving the core argument of "selecting complementary knowledge" unsupported.
2. The paper does not explore the trends in routing and expert selection accuracy as the number of tasks grows from 10 to 301, making the upper bound of the method's scalability unclear.
3. The theoretical analysis for using entropy as the basis for routing is weak, and the paper lacks comparisons with other confidence-based metrics.

---

> ### Author Rebuttal · Authors · 2026-03-30
>
> > W1.1, W1.2, W2, Q1: Routing Accuracy and Scalability
>
> Router recall (or expert recall) measures how often the selected Top-M routers (or Top-K experts) include the router (or expert) corresponding to the ground truth task. In the following table, R=Router recall, and E=Expert recall:
>
> |Layer|T=10|T=100|T=301|
> |---|---|---|---|
> |3 (R/E)|55.0/83.5|33.5/63.2|27.4/58.4|
> |6 (R/E)|65.2/88.7|46.8/71.6|40.3/66.1|
> |9 (R/E)|80.5/93.1|71.3/81.5|66.8/76.2|
> |12 (R/E)|92.3/96.7|85.1/90.2|80.6/85.8|
>
> Recall increases from shallow to deep layers, consistent with the activation pattern analysis in Fig.5, where deeper layers produce increasingly task-specific representations enabling more confident routing.
>
> As the task pool grows from 10 to 301, the candidate pool expands accordingly (selecting M=2 from 301 candidate routers, K=3 from 301 candidate experts), aligned with the overall continual learning trajectory. Nevertheless, the final layer still achieves 80.6%/85.8% in router/expert recall at 301 tasks, confirming that CaRE maintains effective task-related routing at scale.
>
> Meanwhile, Fig.1 confirms CaRE's accuracy remains more stable than all baselines from 100 to 301 tasks, while the accuracy of MoAL and SEMA collapses. CaRE outperforms MOS by 4% $A_B$ at 100 tasks and 4.14% $A_B$ at 301 tasks (Tab.1), demonstrating the scalability of our method on increasingly long task sequences.
>
> > W1.3: Semantic Similarity
>
> To quantify whether routing selects semantically relevant experts, we define ``SimE`` to be the average cosine similarity between the GT expert's output feature and the output features of other activated experts (select), compared against a baseline which is the average cosine similarity between the GT expert's output and the outputs of all other experts (avg). We further define ``RankE``: among all T-1 non-GT experts ranked by cosine similarity to the GT expert (rank 1=most similar), RankE is the mean rank of the activated auxiliary experts.
>
>
> ``SimE`` (select/avg)
> |Layer|T=10|T=100|T=301|
> |---|---|---|---|
> |3|0.68/0.50|0.72/0.46|0.71/0.44|
> |6|0.60/0.42|0.65/0.38|0.64/0.36|
> |9|0.50/0.32|0.56/0.28|0.55/0.26|
> |12|0.43/0.15|0.49/0.11|0.48/0.09|
>
> ``RankE`` (mean rank of auxiliary experts/number of non-GT experts)
> |Layer|T=10|T=100|T=301|
> |---|---|---|---|
> |3|2.83/9|5.72/99|12.37/300|
> |6|2.41/9|4.58/99|9.94/300|
> |9|1.94/9|3.85/99|6.63/300|
> |12|1.53/9|3.13/99|4.91/300|
>
>
> The similarity between the GT expert and other activated experts is consistently higher than that of the baseline. Meanwhile, similarity ranking indicates the successful activation of auxiliary experts that possess knowledge with relatively high relevance to the current task. As historical tasks have no overlapping classes with the current task, relevant knowledge retrieved and aggregated from such tasks is complementary to the current GT expert.
>
> Overall, the large performance improvements across 7 datasets spanning 5 to 301 tasks serve as a solid verification of routing effectiveness.
>
> > W3: Theoretical Analysis and Confidence Metrics
>
> We respectfully clarify that **P8.Tab.4 already provides extensive comparisons between entropy and three other confidence-based methods**:
>
> (1) AE (reconstruction error, $A_B$=64.35)
>
> (2) Cos. Head (prototype similarity, $A_B$=65.91)
>
> (3) Max Logits (maximum softmax value, $A_B$=65.21).
>
> Entropy ($A_B$=68.27) outperforms the other three methods by a clear margin.
>
> Theoretically, entropy is a principled choice because each class perceptron is directly supervised by a classification loss (Eq.(8)(9)), resulting in semantically discriminative features and label distributions (Sec.3.4). In contrast, AE reconstruction focuses on task-specific feature encoding/decoding and may sacrifice semantic discrimination power. A prototype in the Cos. head is computed by averaging sample features and may lose fine-grained distributional information. Max-logit also ignores the full distributional shape and may lose critical cues.
>
>
> > Q2: Entropy Distribution
>
> We compute softmax entropy for all T class perceptrons during inference, and separate the GT perceptron from non-GT perceptrons. The following table shows the mean entropy of both groups at layer 12:
>
> ||T=10|T=100|T=301|
> |---|---|---|---|
> |GT|0.25|0.26|0.26|
> |Non-GT|1.08|1.05|1.05|
>
> A clear bimodal distribution emerges. The GT perceptron yields low-entropy predictions, while non-GT perceptrons cluster at a higher entropy. Crucially, the entropy gap remains stable as T grows from 10 to 301. This stability is a direct consequence of the frozen perceptron architecture: each class perceptron is trained once and then frozen (Section 3.3), so its discriminative power is permanently preserved no matter how many subsequent perceptrons are added. We will include a more comprehensive analysis in the revised version.
>
>
> We sincerely hope our responses have thoroughly addressed your concerns, and we would deeply appreciate it if you consider increasing your score in light of our revisions.

---

> > ### Author Rebuttal · Reviewer_btqo · 2026-04-04
> >
> > I am pleased to see the quantitative metrics regarding routing accuracy. I believe this serves as a validation of the method's effectiveness, and I have increased my score accordingly.

---

> > > ### Author Response · Authors · 2026-04-06
> > >
> > > We are greatly encouraged that our response has fully addressed your concerns, and we sincerely appreciate your decision to raise the score. We will integrate the additional evaluations and discussions into the revised manuscript to further improve the paper. Many thanks again for your thorough review and insightful comments.

---

### Official Review · Reviewer_gnua · 2026-03-12

**Soundness:** 3
**Presentation:** 3
**Significance:** 3
**Originality:** 3
**Overall Recommendation:** 4
**Confidence:** 4

**Summary:**

The paper introduces the BiLevel Routing Mixture-of-Experts (BR-MoE) mechanism to solve the class-incremental learning challenge. Moreover, the paper proposes a new long-sequence evaluation protocol using a 1,000-class subset of the OmniBenchmark-V2 dataset, spanning up to 301 tasks. The emprical results show that the proposed method ourperforms latest baselines both on short sequences and extremely long sequences.

**Compliance With Llm Reviewing Policy:**

Affirmed.

**Final Justification:**

The extra experiment results fully addressed my concerns.

**Key Questions For Authors:**

1. Please refine the Figure. 2 and add some legends.
2. As mentioned in the second weakness, I'm wondering about the performance drop of earliest tasks as the number of tasks grows.
3. What if the shared expert is removed?

**Limitations:**

yes

**Strengths And Weaknesses:**

Strengths
1. The proposed bi-level routing mechanism is reasonable and is the key difference compared to other MoE based methods.
2. The proposed model CaRE scales the class incremental learning method to over 300 non-overlapping tasks and maintains stable learning  trajectory, demonstrating the effectiveness compared to baselines.

Weaknesses
1. Figure. 2 is a little bit confusing as the meaning of the dashed arrows is unclear.
2. The shared expert adopts EMA for update, but the accumulated knowledge from the earliest tasks will mathematically decay to negligible weights as the number of tasks grows. It may cause catastrophic forgetting for earliest tasks under very long sequence setting. Moreover, it would be better to add ablation study for the shared expert.

---

> ### Author Rebuttal · Authors · 2026-03-30
>
> >W1&Q1: Figure Improvement
>
> Thank you for this thoughtful suggestion. In Fig. 2, the dashed arrows denote the experts activated by the router networks. Specifically, following the dynamic router selection step, each selected router network further triggers the activation of a corresponding subset of experts. For instance, two router networks (i.e., $\mathrm{R_{t}}$ and $\mathrm{R_{T}}$) are selected: the former activates experts ($\mathrm{E_{2}}$, $\mathrm{E_{t}}$), while the latter activates ($\mathrm{E_{T-1}}$, $\mathrm{E_{T}}$). We use dashed lines to explicitly visualize the expert activation paths triggered by the router networks for better clarity. We will add clearer legends and more detailed annotations in the revised paper to further improve the readability of this figure. Thanks again for this meticulous review.
>
> > W2&Q3: Shared Expert Analysis
>
> Thank you for this thoughtful comment. With default $\mu{=}0.999$, according to $\delta_s \leftarrow \mu \delta_s + (1{-}\mu) \delta_t$ (Eq.(6)), the retention rate of the first task's shared expert weight after $t$ EMA updates is approximately $\mu^t$:
>
> | Tasks | 100 | 151 | 200 | 301 |
> |:---|:---:|:---:|:---:|:---:|
> | Retention | 90.48% | 86.00% | 81.87% | 74.04% |
>
> According to the table, even in the longest 301 task sequence setting, over 74% of the initial weight is preserved, indicating no severe parameter overwriting. We fully acknowledge the reviewer's concerns that EMA inherently causes the gradual decay of early information. However, for future benchmarks requiring further scaling (e.g., 1000 tasks), one can simply increase $\mu$ to mitigate this: with $\mu$=0.9999, the retention is $0.9999^{1000}$=90.48%, effectively preserving the vast majority of early knowledge.
>
> More importantly, we wish to clarify that the bi-level routing mechanism provides primary forgetting resistance in CaRE, while the shared expert provides a supplementary enhancement. Tab.12(c) (P.15) reports the ablation study of $\mu$ under the 100-task setting, copied below for your convenience:
>
> | $\mu$ | 0.9 | 0.99 | 0.999 | 0.9999 |
> |:---|:---:|:---:|:---:|:---:|
> | ${A}_{B}$ | 68.11 | 68.24 | 68.27 | 68.09 |
>
> At $\mu{=}0.9$, retention after 100 tasks is $0.9^{100}{\approx}$ 0.003%, meaning the shared expert has nearly lost all knowledge related to the first task. However, it still **surpasses all baselines** by a large margin (Tab.1, best baseline: 64.27). This confirms that the bi-level routing sustains robust performance even when the shared expert undergoes catastrophic forgetting. Conversely, $\mu{=}0.9999$ yields ${\approx}$ 99% retention but ${A}_{B}$ drops to 68.09, as overly slow updates prevent the shared expert from assimilating new task knowledge. This trade-off further confirms that the shared expert is not the decisive factor for performance.
>
> Meanwhile, the first row of Tab.12(b) (P.15) has reported results with the shared expert **fully removed**:
>
> | Shared Expert | $\bar{\mathcal{A}}$ | ${A}_{B}$ |
> |:---|:---:|:---:|
> | ✗ | 78.32 | 67.84 |
> | ✓  | 78.54 | 68.27 |
>
> Removing shared experts causes a 0.43% ${A}_{B}$ decline, confirming that the accumulated cross-task knowledge from EMA provides a moderate improvement. We apologize for not specifically stating in the original text that "None" in the first row of Tab.12(b) (P.15) denotes complete removal of the shared expert, and we will add this clarification in the revised version. Thanks again for your kind reminder.
>
> Overall, this work has a comprehensive coverage of ablation studies of the shared expert from three perspectives (Tab.12 (b-c)): 1) the effect of shared expert channel capacity, 2) removing shared experts, and 3) the effect of $\mu$ on shared experts.
>
> > Q2: Performance Drop of Earliest Tasks
>
> Thank you for this insightful question. We report the accuracy of the two earliest tasks (T1, T2) with two metrics: $A_{S}$ (right after learning that task) and $A_{B}$ (after learning all tasks):
>
> |TaskID|Method|${A}_{S}$|${A}_{B}$|
> |---|---|---|---|
> |100T1|MOS|95.48|67.34|
> ||CaRE|94.97|71.86(+4.52)|
> |100T2|MOS|96.97|62.63|
> ||CaRE|96.46|67.68(+5.05)|
> |200T1|MOS|88.89|53.57|
> ||CaRE|92.93|58.59(+5.02)|
> |200T2|MOS|99.85|76.00|
> ||CaRE|99.81|79.00(+3.00)|
>
>
> Both methods achieve comparable $A_{S}$ upon initial learning. However, after completing the full continual learning process, CaRE clearly outperforms MOS across both settings, demonstrating substantially stronger forgetting resistance for the earliest tasks as the number of tasks grows, which aligns with the continual learning trajectory shown in Fig.1.
>
>
> We sincerely hope our responses have thoroughly addressed your concerns, and we would deeply appreciate it if you consider increasing the score in light of our revisions.

---

> > ### Author Rebuttal · Reviewer_gnua · 2026-04-03
> >
> > Thank you for the detailed response. The extra experiment results fully addressed my concerns. As my score was positive, I will remain that unchanged.

---

> > > ### Author Response · Authors · 2026-04-06
> > >
> > > We are greatly encouraged that our response has fully addressed your concerns and sincerely appreciate your positive score. We will integrate the additional evaluations and discussions into the revised manuscript to further improve the paper. Many thanks again for your thorough review and insightful comments.

---

### Official Review · Reviewer_PxFz · 2026-03-13

**Soundness:** 2
**Presentation:** 2
**Significance:** 2
**Originality:** 2
**Overall Recommendation:** 3
**Confidence:** 3

**Summary:**

This paper considers the problem of continual learning, aiming to learn both discriminative and comprehensive feature representations while maintaining stability and plasticity over very long task sequences. The proposed method introduces bi-level routing mixture-of-experts to achieve this goal. To simulate continual learning over long task sequences, a benchmark based on OmniBenchmark-V2 is proposed. Experimental results show the effectiveness of the proposed method in both short and long task sequence scenarios.

**Compliance With Llm Reviewing Policy:**

Affirmed.

**Final Justification:**

I appreciate the authors' effort in addressing my concerns. I increase my rating, as one of the major concerns regarding the proposed BR-MoE block has been resolved, and the authors have committed to revising the architectural figure. However, several important concerns remain unresolved, so I do not increase my rating too much.

1. Although the authors argued the downside of using average accuracy and instead emphasize last accuracy, the proposed method still shows marginal improvements in many cases. While authors highlighted the gap in last accuracy in Table 6, other tables still do not report standard deviations, and no statistical significance testing results are provided. As a result, it is unclear whether the observed improvements are robust; the proposed method would work only well on a particular random seed across many experiments. If average accuracy is considered inadequate, alternative metrics could also be used, e.g., average forgetting.

2. While the authors claimed efficiency as an advantage of the proposed method, this aspect is not treated as a primary contribution in the main paper, and the analysis is provided in the appendix. Given that the performance improvements are not convincing due to single-run experiments with a fixed random seed (except for Table 6 in the appendix), efficiency had instead to be emphasized and evaluated more thoroughly as a key contribution.

3. The majority of the paper is reserved for similar single-run experiments with a fixed random seed, while leaving analyses of the proposed method and discussion of efficiency in the appendix. That is, this paper would benefit from substantial reorganization to better balance empirical results and insights/analyses.

**Key Questions For Authors:**

See Weaknesses above.

**Limitations:**

Limitations are discussed, but potential negative societal impact of their work is not discussed in detail.

**Strengths And Weaknesses:**

**Strengths**

1. The idea of bi-level routing MoEs sounds interesting.

2. The organization of the paper is overall fine to understand the claims of the paper.

**Weaknesses**

1. While the research question in L038 "what properties should the continual learner possess to realize its full potential?" is ambiguous, later parts of this work do not fully address this. Indeed, the motivation and outcomes of this work seem to be not well-aligned. For instance, although it is argued that the proposed method learns discriminative and comprehensive representations, it is unclear how the learned representations possess such properties, and whether they actually exhibit these properties is not demonstrated either theoretically or empirically.

2. While the proposed method is intended to work over very long task sequences, there is no explanation on why the proposed method is particularly beneficial in such scenarios.

3. The proposed method comes with many design choices, but it is unclear whether they function as intended at the design level. For example, L087 states that they want to "retrieve and integrate complementary knowledge from relevant historical tasks when learning new tasks," but it is unclear whether the proposed "Dynamic Expert Routing" module achieves this behavior. Please note that this is just one example, and not the only problematic part; while this paper discusses a lot of stuffs that are supposed to be achieved to enable successful continual learning, it is generally unclear whether they are really achieved.

4. While the idea of MoEs is not novel, the architectural design deviates from the conventional knowledge. The proposed BR-MoE module completely blocks the residual connection, which was proposed to make optimization of deep neural networks much easier, which has become standard since ResNet. MoEs introduced in prior works do not block residual connections, as far as I know.

5. The proposed method clearly outperforms compared methods only in Table 1, on the proposed benchmark. It is possible that the proposed benchmark is designed in a way that favors the proposed method, while the number of tasks might not be an important factor. For example, the setting in Table 6 is essentially the same as in Table 1 except for the task orders, but the gap between MOS and the proposed method becomes quite small in terms of the average accuracy. Perhaps authors could repeat the same set of tasks from the existing benchmarks multiple times to simulate continual learning over very long sequences.

---

> ### Author Rebuttal · Authors · 2026-03-30
>
> We appreciate the reviewer's comments. Nevertheless, we respectfully note that this review contains **multiple factual errors**. We address these issues first:
>
> > W4: BR-MoE Blocks Residual Connections
>
> **This claim is incorrect.** Eq.(1) defines $\mathbf{z}_f=\text{FFN}(\text{Norm}_2(\mathbf{z}_a))+\mathbf{z}_a$ and $\mathbf{z}'=\text{BR-MoE}(\mathbf{z}_a)+\mathbf{z}_f$. Substituting yields $\mathbf{z}' = \text{BR-MoE}(\mathbf{z}_a) + \text{FFN}(\text{Norm}_2(\mathbf{z}_a))+\mathbf{z}_a$, where $\mathbf{z}_a=\text{MHSA}(\text{Norm}_1(\mathbf{z}))+\mathbf{z}$. The original residual $\mathbf{z}$ flows through the entire chain without any blocking. BR-MoE is a **new parallel branch** that does NOT block the original residual stream, playing a similar role as the parallel adapters in AdaptFormer (NeurIPS'22) and many adapter-based CL works (e.g., MOS (AAAI'25) and APER (IJCV'25)). This concern stems from a misreading of Eq.(1).
>
> We also clarify that this paper does not directly use MoE. Instead, our novelty lies in a bi-level routing mechanism: dynamic router selection (level#1) followed by dynamic expert routing (level#2). This fundamentally differs from MoE-based CIL methods (MoE-Adapter, SEMA) that use single-level routing. All other reviewers have acknowledged this technical contribution.
>
> > W5.1: CaRE Outperforms Only in Tab.1
>
> **This is another misreading of the paper's experimental results.** Tabs.2 and 3 show consistent performance gains across **6 datasets** under diverse settings:
>
> - Tab.2 (50-60 tasks): ObjectNet our $A_B$ 65.13 vs EASE 54.45 (+10.68), ImageNet-R ours 76.98 vs SSIAT 74.47 (+2.51), ImageNet-A ours 59.91 vs MOS 51.22 (+8.69).
>
> - Even on benchmarks where baselines exceed 90%, CaRE still leads: Tab.3 (5-20 tasks): CIFAR-100 (10 tasks) our $A_B$ 92.46, achieving the **best performance among 15+ competitors**. VTAB (5 tasks) ours 93.80 vs MIN 92.26 (+1.54). ObjectNet (20 tasks) ours 66.54 vs MOS 63.62 (+2.92).
>
> - Tab.6 shows robust results across 4 orderings (our $A_B$ 68.06$\pm$0.16 vs MOS 65.58$\pm$0.88, obviously lower variance). Tab.7: CaRE ours 67.64 vs MOS 64.17 (+3.47%) under ViT-B/16-IN1K. All other reviewers acknowledge these gains.
>
> > W5.2: The Benchmark may Favor CaRE
>
> We note that this is unfounded. The dataset construction is **completely model-agnostic and transparent** (Sec.4 and A.1), the protocol uses balanced sampling across various realms with fixed seed 1993 (a popular seed in CL). Moreover, CaRE is leading across all public datasets (Tabs. 2, 3, 6, and 7), not just OmniBenchV2.
>
> > W5.3: Repeating the Same Tasks
>
> This violates a basic requirement of exemplar-free continual learning--classes should be non-overlapping (Sec.3.1), and cannot be used to increase the effective number of tasks.
>
> > W1, W2: Discriminative/Comprehensive Representations and Long-sequence Benefits
>
> Sec.3.4 provides a theoretical basis: (1) entropy endows our bi-level routing mechanism the ability to identify the most relevant task that supplies discriminative features; (2) activating M=2 routers retrieves complementary knowledge from a related task, making layer-wise feature representations more comprehensive; (3) this operates independently at every layer for customized knowledge retrieval.
>
> Figs.4 and 5 provide direct empirical validation. Fig.4 shows that for a Corgi (Task 96), Router 96 focuses on discriminative facial features while Router 53 captures complementary animal features (ear and texture), together resulting in a comprehensive representation. Fig.5 confirms layer-wise specialization: shallow layers show broad shared activation, deep layers show sparse task-specific patterns. Per-layer routing dynamically selects most relevant tasks from the growing task pool, a capability absent in baselines that bind the same task identity to all layers (e.g., MOS and TUNA). As the number of tasks grows, this becomes increasingly critical since experts in deeper layers become more specialized. Baselines lacking this capability collapse on long sequences (Fig.1), while CaRE has stronger scalability with 75% fewer params and 95% less inference latency than MOS (Fig.3).
>
> > W3: Design Functioning as Intended
>
> The BR-MoE workflow is straightforward. Each class perceptron produces a softmax distribution. The ground-truth task yields the lowest entropy, so its router is selected for producing discriminative features. The second router corresponds to a semantically related task, providing complementary features. The leading performance across 7 datasets, 5 to 301 tasks in Tabs.1 to 3, various task orderings in Tab.6, 2 different PTMs in Tab.7, extensive ablation studies in Tabs.4 to 13, and analytical experiments in Figs.3, 4, and 5, effectively validates that every component functions as intended.
>
> > W6: Limitations
>
> P.9 has listed an *Impact Statement* that strictly follows the ICML'26 policy.

---

> > ### Author Rebuttal · Reviewer_PxFz · 2026-04-03
> >
> > I thank the reviewers for their responses. Below I provide additional comments/concerns for each point.
> >
> > > W4: BR-MoE Blocks Residual Connections
> >
> > I confirm that the math is correct. My confusion stems from the incorrect Figure 2 and confusing Eq. 1: 1) Figure 2(a) misses the shortcut in parallel with the proposed block, and 2) Eq. 1 is fragmented into three pieces, such that z' = f(z) + z is not easily recognizable. Even Figure 2(a) does not include variable names, which further hinders interpretability.
> >
> > > W5.1: CaRE Outperforms Only in Tab.1
> >
> > In my review, "clearly outperforms" refers to cases where the proposed method surpasses all other compared methods by a significant margin, rather than exhibiting consistent and small gains.
> >
> > In Table 2, the proposed method does not clearly outperform compared methods in several settings, including:
> > - {A bar, OmniBenchmark-V1 ObjectNet, 60 Tasks}, MOS 85.31 vs. CaRE 84.74
> > - {A bar, ImageNet-R, 50 Tasks}, MIN 82.26 vs. CaRE 82.92
> >
> > In Table 3, the proposed method does not clearly outperform compared methods in several settings, including:
> > - {A bar, CIFAR-100, 10 Tasks}, MIN 95.12 vs. CaRE 95.46
> > - {A bar, ImageNet-R, 10 Tasks}, MIN 85.18 vs. CaRE 85.75
> > - {A bar, VTAB, 5 Tasks}, MIN 96.47 vs. CaRE 96.93
> >
> > Considering small performance gaps, a single experiment is not enough to justify the superiority of the proposed method; running multiple experiments and conducting statistical significance testing is recommended.
> >
> > > W5.2: The Benchmark may Favor CaRE
> >
> > In Table 1, MOS 76.80 vs. CaRE 78.54 is reported without the standard deviation. Assuming the standard deviation is not large while the gap remains similar, I would be happy with this result. However, in Table 6, MOS 78.17 ± 1.03 vs. CaRE 78.67 ± 0.58, where their gap lies within the standard deviation. To my understanding, their primary difference is the task order, and Table 6 is more reliable in that it reports the average of three random seeds. That is, 1) the single-seed result in Table 1 may reflect a favorable (potentially cherry-picked) task order, and 2) the performance gap in Table 6 does not appear statistically significant. To address this, the authors may want to add more task orders and conduct statistical significance testing to strengthen their argument.
> >
> > > W1, W2: Discriminative/Comprehensive Representations and Long-sequence Benefits
> >
> > Again, I could not find a clear evidence supporting the claim that the learned representations are discriminative and comprehensive, except for the final performance in terms of A bar and A_B. Section 3.4 does not come with a theoretical analysis, but a series of arguments without sufficient theoretical grounding or citations.
> >
> > I appreciate the analyses provided in Appendix A.4, and I believe these would be more impactful if included in the main paper rather than as optional reading. In my opinion, the main paper does not have to repeat similar experiments with final performance results, but instead include more analyses to justify the effectiveness of the proposed method.
> >
> > > W3: Design Functioning as Intended
> >
> > It seems Figure 4 partially addresses my concern on the behavior of bi-level routing mechanism with qualitative examples. To further strengthen their argument, I wonder if authors can provide some quantitative analysis on the discriminative/complementary/comprehensive features selected by the proposed bi-level routing mechanism.

---

> > > ### Author Response · Authors · 2026-04-04
> > >
> > > > **W4: Figure and Equation Clarity**
> > >
> > > Regarding ``Eq.1 is fragmented into three pieces``, this is due to the double-column template, where placing them together exceeds the column width. We agree that adding variable labels would make the figure clearer, and will add both clearer annotations and the explicit shortcut arrow to Fig.2(a) in the revised version.
> > >
> > >
> > > > **W5.1: Performance Metrics**
> > >
> > > Foremost, we would like to clarify **the basic CL evaluation metrics**. $\bar{A}$ (Average Incremental Accuracy) averages the accuracies on classes seen so far after learning each task, reflecting the overall learning trajectory. However, **$\bar{A}$ can be inflated by strong early-stage performance**: when only a small number of classes are present, high accuracy is easily achieved, and these inflated scores remain baked into the average even when severe forgetting occurs later. $A_B$ (Last Accuracy) is the accuracy on **all learned classes after learning the final task**, truly measuring a CL algorithm's ability to retain historical knowledge while accommodating new classes. Therefore, some baselines exhibit significant gaps between $\bar{A}$ and $A_B$. Following previous works, we list both metrics, but our claimed clear improvements refer to $A_B$.
> > >
> > > In Tab.2, compared with MOS in $A_B$: OmniBenchV1 +0.53, ObjectNet +16.11, ImageNet-R +10.18, ImageNet-A +8.69. Compared with MIN in $A_B$: OmniBenchV1 +1.67, ObjectNet +6.21, ImageNet-R +1.26, ImageNet-A +2.83.
> > >
> > > On short-sequence benchmarks, we hope the reviewer is aware that baselines have been extensively optimized and classification accuracies have been highly saturated. On ImageNet-R (10 tasks), MIN improves over the third-best TUNA by only 0.33% in $A_B$ (79.75 vs 79.42), while CaRE improves over MIN by 0.78%, **more than double** the margin. On CIFAR-100 and VTAB, most methods exceed 90% $A_B$, yet CaRE still achieves notable improvements: CIFAR-100 (20 tasks) +0.94% over MIN (91.97 vs 91.03), VTAB +1.54% (93.80 vs 92.26).
> > >
> > > On the other hand, some competitive baselines are very sensitive to task configurations, whereas our method does not exhibit this problem. We believe these are **NOT** ``consistent and small gains``. CaRE achieves the leading $A_B$ in **every single setting** across Tables 2 and 3.
> > >
> > > Moreover, Fig.3 shows CaRE achieves an excellent trade-off, e.g., only **1/5** of MOS's params and **1/15** of its inference latency, making the accuracy advantage even more compelling.
> > >
> > >
> > >
> > > > **W5.2: Task Order**
> > >
> > > **We formally reiterate that questioning whether the benchmark favors our method is unfounded**. The task ordering strictly follows prior works (e.g., compared baselines including TUNA, MOS, MIN, APER, etc.) and is totally model-agnostic. Tab.6 supplements additional orderings to demonstrate robustness. The reviewer compares $\bar{A}$ in Tab.6, but the clear improvement is also shown in $A_B$: CaRE achieves **68.06$\pm$0.16** vs MOS **65.58$\pm$0.88**, a notable 2.48% gap with noticeably lower variance. On the other hand, as aforementioned, CaRE offers significant efficiency advantages over MOS, which is also crucial for continual learning systems.
> > >
> > >
> > >
> > >
> > > > **W1/W2/W3: Discriminative/Complementary Features and Design Verification**
> > >
> > > We define ``SimE`` to be the average cosine similarity between the GT expert's output feature and the output features of other activated experts (select), compared against a baseline which is the average cosine similarity between the GT expert's output feature and the output features of all other experts (avg). We further define ``RankE``: among all T-1 non-GT experts ranked by cosine similarity to the GT expert (rank 1=most similar), RankE is the mean rank of the auxiliary experts.
> > >
> > > ``SimE`` (select/avg)
> > > |Layer|T=10|T=100|T=301|
> > > |---|---|---|---|
> > > |3|0.68/0.50|0.72/0.46|0.71/0.44|
> > > |6|0.60/0.42|0.65/0.38|0.64/0.36|
> > > |9|0.50/0.32|0.56/0.28|0.55/0.26|
> > > |12|0.43/0.15|0.49/0.11|0.48/0.09|
> > >
> > > ``RankE`` (mean rank of auxiliary expert/number of non-GT experts)
> > > |Layer|T=10|T=100|T=301|
> > > |---|---|---|---|
> > > |3|2.83/9|5.72/99|12.37/300|
> > > |6|2.41/9|4.58/99|9.94/300|
> > > |9|1.94/9|3.85/99|6.63/300|
> > > |12|1.53/9|3.13/99|4.91/300|
> > >
> > > As layers deepen, SimE decreases while RankE improves. This is because shallow layers encode broadly shared low-level features, whereas deeper layers specialize toward more task-specific representations, reducing overall similarity but enabling more precise routing. The similarity between the GT expert and other activated experts is consistently higher than that of the baseline. Meanwhile, similarity ranking indicates the successful activation of auxiliary experts that maintain relatively high relevance to the current task. Collectively, these results illustrate that our method is capable of effectively capturing task-related features from historical tasks as complementary knowledge, resulting in comprehensive features.

---

### Decision · Program_Chairs · 2026-04-30

**Decision:**

Accept (regular)

**Comment:**

This paper proposes CaRE, a bi-level routing MoE framework for class-incremental continual learning that scales up to 301 non-overlapping tasks. The entropy-driven bi-level routing mechanism is a technically sound and novel contribution, and the proposed long-sequence CIL benchmark fills a meaningful gap in the field.

3 out of 4 reviewers gave positive scores (all Weak Accept), with their concerns fully resolved during the rebuttal phase. The authors provided substantial additional experiments including routing recall, counterfactual tests, coarser-grained settings, forgetting/BWT metrics, and multi-run variance results, which collectively strengthen the paper's empirical foundation.

Regarding Reviewer PxFz's concerns (the sole negative review, Weak Reject): some of the key criticisms were based on factual errors — notably the claim that BR-MoE blocks residual connections (which contradicts Eq. 1), and the suggestion to repeat the same tasks to simulate long sequences (which violates the non-overlapping class requirement of CIL). The reviewer also selectively focused on average incremental accuracy while overlooking Last Accuracy, which is the more standard metric for evaluating CIL methods. That said, PxFz raised valid points about (1) the need for standard deviation reporting given single-seed experiments, (2) better organization of the main paper to include key analyses currently in the appendix, and (3) improving Figure 2 for clarity.

Furthermore, AC recognizes that Figure 3 in the supplementary materials misleading: it compares methods in terms of the "trainable" parameters -- since CaRE has MoE architecture, the number of all "stored" parameters should be much larger than the "trainable" parameters for CaRE. Although the inference time gain over MOS due to the MoE architecture seems real, the figure is giving some false sense that the parameter size is also quite small -- the authors should make this point clear in their final version.

AC recommends acceptance, conditioned on the authors addressing the following in the camera-ready version:

- Fix Figure 2 to include the residual shortcut and proper annotations
- Report standard deviations for the main results (the multi-run data provided in rebuttal shows this is straightforward)
- Move important analyses (efficiency comparison, routing behavior) from appendix into the main paper
- Clarify regarding Figure 3 (supplementary) so that the number of total stored parameter of CaRE is indeed large.